# How to Teach Label to Understand Decisions: A Decision-aware Label Distribution Learning Framework

## Abstract

Contextual Stochastic Optimization (CSO) aims to predict uncertain, context-dependent parameters to inform downstream decisions. A central challenge is that high predictive accuracy does not necessarily translate into optimal decisions. Existing approaches typically rely on custom loss functions, but these often suffer from non-differentiability, discontinuity, and limited modularity. To address these limitations, we propose a decision-aware Label Distribution Learning (LDL) framework that retains standard loss functions to avoid computational issues, while encoding decision knowledge entirely at the level of data representation. Our approach models uncertainty as full label distributions and reshapes them during the label enhancement stage to reduce predictive mass in high-risk regions. Scalar targets are transformed into individualized mixture distributions using decision-aware similarity matrices, and a dual-branch neural network is trained to learn decision-aware label distributions. Extensive experiments on synthetic benchmarks (e.g., newsvendor, network flow) and real-world datasets demonstrate consistent regret reduction across different sample sizes, with particularly strong improvements in low-data regimes. These results highlight LDL as a promising new pathway for achieving robust and principled decision-making under complex cost structures.

## 1 Introduction

Predict-then-optimize is a widely used paradigm for solving optimization problems under uncertainty. In this framework, given covariates, a contextual predictor first estimates the distribution of the uncertain parameters, and the resulting estimates serve as input to a Contextual Stochastic Optimization (CSO) model (Sadana et al., 2025). The traditional sequential learning-then-optimization (SLO) approach trains the contextual predictor by minimizing an estimation error between the true conditional distribution and the conditional distribution given by the contextual predictor. While effective for improving prediction accuracy, this approach neglects the downstream optimization objective and can therefore result in suboptimal decisions.

To bridge this gap, Integrated Learning and Optimization (ILO) has emerged as a promising alternative (Sadana et al., 2025). ILO methods train contextual predictors explicitly incorporating the downstream decision objective into the learning process, thereby aligning prediction and optimization. The predominant way to realize this is by designing decision-aware loss functions, which maximize decision quality on the training set (Mandi et al., 2024), rather than minimizing an estimation error.

Nevertheless, existing methods are subject to two fundamental limitations. The first concerns the high training cost of loss-function-based approaches. These methods design decision-aware loss functions (e.g., regret) to align predictions with downstream decision quality. However, such losses are often discontinuous and non-differentiable, which makes gradient-based optimization unstable and computationally expensive. Although surrogate losses have been proposed to mitigate this issue (Elmachtoub & Grigas, 2022), they still impose substantially higher training costs than conventional predictive models and frequently rely on task-specific approximations, thereby limiting their general applicability.

The second challenge is the lack of a general and adaptive framework for modeling uncertainty distributions in CSO. Modeling uncertain parameters as continuous distributions often renders the downstream optimization problem intractable, due to the curse of dimensionality arising from high-dimensional integration. A common workaround is to approximate the uncertainty using a discrete distribution. However, most prior work either fixes the discrete support set a priori (Qi et al., 2023), as discussed in Appendix I or derives it solely from the feature space (Bertsimas & Kallus, 2020). Such approaches overlook the fact that the choice of support set itself can have a substantial impact on decision quality.

To address the aforementioned challenges, we introduce the Label Distribution Learning (LDL) framework into CSO. LDL provides a refined way to represent uncertainty through label distributions: point labels are first enhanced into distributional labels, which then serve as the foundation for training conditional distribution predictors (Geng, 2016). This framework not only offers a more flexible mechanism for modeling uncertainty distributions, but also opens up a novel pathway to achieve decision-awareness without relying on loss-function-based methods. In particular, our paper makes the following key contributions:

- **Decision-awareness through label enhancement.** We incorporate decision-awareness at the label enhancement (LE) stage within the LDL framework. This avoids the discontinuity and computational cost associated with decision-aware loss functions while still aligning prediction with downstream decision-making.

- **General and adaptive distribution construction.** We present a method for constructing discrete uncertainty distributions by leveraging the similarity between the feature space and the decision space to determine the support set. Unlike existing methods that fix the support set a priori or derive it solely from features, our approach adapts flexibly across diverse problem settings.

- **Robustness and scalability.** Through extensive experiments on both synthetic and real-world datasets, we demonstrate that our approach effectively reduces decision regret compared to baseline models, leading to more robust and reliable outcomes across diverse problem settings.

## 2 RELATED WORKS

### 2.1 CONTEXTUAL STOCHASTIC OPTIMIZATION

Stochastic optimization is a classical paradigm for decision-making under uncertainty. A common approach is sample average approximation (SAA) (Kleywegt et al., 2002), which replaces the true distribution with an empirical one but ignores covariates. CSO addresses this by leveraging covariates to predict uncertain parameters (Sadana et al., 2025). Within CSO, prescriptive analytics extends SAA by assigning covariate-based weights to samples via k-nearest neighbors, kernel methods, or tree models (Bertsimas & Kallus, 2020), though this SLO method can yield suboptimal decisions.

To overcome this, ILO methods jointly train predictive models and decision tasks, typically through customized decision-aware loss functions. However, such losses are often discontinuous and non-differentiable, hindering gradient-based training (Mandi et al., 2024). Solutions include surrogate-based methods such as SPO+ for linear objectives (Elmachtoub & Grigas, 2022), conditional estimation–optimization (ICEO) for discrete distributions (Qi et al., 2023), perturbed maximizers (Berthet et al., 2020), differentiable solver modules (Sahoo et al., 2023; Vlastelica et al., 2020), and gradient-free models like decision trees with decision-aware objectives (Elmachtoub et al., 2020; Kallus & Mao, 2023).

Unlike prior work centered on loss design, our approach embeds decision-awareness during the LE stage within the LDL framework. This avoids reliance on differentiable surrogates or gradient propagation from the optimization model, thereby sidestepping limitations of traditional decision-focused learning.

## 2.2 LABEL DISTRIBUTION LEARNING

LDL addresses the ambiguity in real-world labeling by assigning each instance a distribution of description degrees across labels. Unlike single-label learning, which fixes a definitive label, or multi-label learning, which uses binary indicators without graded relevance, LDL represents supervision as a probability-like vector summing to one, thereby quantifying relative importance (Geng, 2016). Its foundations draw on fuzzy logic and probabilistic labeling, formalized as learning a conditional probability mass function to minimize divergences such as Kullback-Leibler. Early methods included problem transformation (e.g., PT-Bayes, PT-SVM), algorithm adaptation (e.g., AA-kNN via neighbor averaging, AA-BP with softmax), and specialized algorithms (e.g., SA-IIS, SA-BFGS) (Zheng et al., 2018). Evaluations across yeast gene expression, natural scenes, and facial datasets (SJAFFE, SBU-3DFE) employed diverse metrics (Chebyshev, Clark, Canberra, KL, cosine, intersection), where specialized designs often performed best (Jia et al., 2018).

To address data scarcity, LE reconstructs distributions from logical labels, with Graph Laplacian LE (GLLE) exploiting topology and correlations (Xu et al., 2021; Gu et al., 2025). Integrated approaches like Directly LDL jointly optimize LE and LDL via KL-divergence and alternating optimization, supported by Rademacher bounds and strong benchmarks (Jia et al., 2023). Objective mismatches are alleviated by Label Distribution Learning Machine (LDLM), which extends margins with SVR and adaptive losses, achieving top performance in 76.5% of tasks (Zhao et al., 2023). For ordinal data, Ordinal LDL applies sequential objectives such as Cumulative Absolute Distance, Quadratic Form Distance, and Cumulative Jensen-Shannon, yielding significant gains in age, beauty, and acne grading (Wen et al., 2023).

By representing supervision as distributions, LDL captures label ambiguity and relative importance beyond traditional settings. When applied to CSO, it enables encoding uncertainty directly in prediction, avoiding discontinuous decision-aware losses. Decision-awareness is embedded during label construction and enhancement, aligning predictive distributions with downstream optimization and enhancing robustness in decision quality—forming the basis of our proposed decision-aware LDL framework.

## 3 PROBLEM STATEMENT

In CSO, the decision-maker selects a decision variable $\mathbf{z} \in \mathcal{Z}$ to minimize the expected task cost under uncertain parameters:

$$\mathbf{z}^*(\mathbf{x}) = \arg\min_{\mathbf{z} \in \mathcal{Z}} \mathbb{E}_{\mathbf{y} \sim P(\mathbf{y}|\mathbf{x})} \left[ c(\mathbf{z}, \mathbf{y}) \right], \tag{1}$$

where $\mathbf{x} \in \mathcal{X}$ is the observed context, $\mathbf{y} \in \mathcal{Y}$ represents uncertain problem parameters, and $c(\mathbf{z}, \mathbf{y})$ is the task-specific cost function. A fundamental challenge arises because the conditional distribution $P(\mathbf{y} \mid \mathbf{x})$ is unknown in practice. Here, we approximate this distribution using a parameterized predictor $f(\cdot; \theta)$ parameterized by $\theta$, taking $\mathbf{x}$ as input and outputting the corresponding distribution over $\mathbf{y}$.

The contextual predictor is typically learned from historical data. It is important to note that data on the conditional distribution $P(\mathbf{y} \mid \mathbf{x})$ is often unavailable. Instead, we have a training dataset $\mathcal{D} = \{(\mathbf{x}_i, \mathbf{y}_i)\}_{i=1}^{N}$. The problem of interest is how to train such a predictor $f(\cdot; \theta)$ so that the resulting decisions $\mathbf{z}^*(\mathbf{x})$ yield low expected cost in the downstream optimization task.

## 4 METHODOLOGY

This section introduces the decision-aware LDL pipeline, which constructs enhanced label distributions from feature and task information and trains a model to predict these distributions for downstream decision-making.

## 4.1 DECISION-AWARE LEARNING AND DECISION-MAKING PIPELINE WITH LABEL DISTRIBUTIONS

LDL first transforms each target into a distribution to capture its uncertainty in LE stage, and then learns a predictive model to map features to these distributions. Figure 1 illustrates the overall structure of the framework. The pipeline consists of two stages:

- **Label Enhancement:** Transform the regression dataset $\mathcal{D} = \{(\mathbf{x}_i, \mathbf{y}_i)\}_{i=1}^{N}$, where $\mathbf{y}_i = (y_i^{(1)}, \ldots, y_i^{(K)})$ denotes the $K$ uncertain parameters for sample $i$, into an enhanced dataset $\mathcal{D}' = \{(\mathbf{x}_i, p_i(\mathbf{y}))\}_{i=1}^{N}$, where $p_i(\mathbf{y}) = \prod_{k=1}^{K} p_i(y^{(k)})$ represents the joint distribution composed of the marginal distributions $p_i(y^{(k)})$.

- **Label Distribution Learning:** Learn a vector-valued function $f(\cdot; \theta) = (f_1(\cdot; \theta_1), \ldots, f_K(\cdot; \theta_K))$, where each component $f_k(\mathbf{x}_i; \theta_k)$ predicts the marginal distribution $p_i(y^{(k)})$, and the joint distribution is reconstructed as $p_i(\mathbf{y}) = \prod_{k=1}^{K} f_k(\mathbf{x}_i; \theta_k)$ optimized for downstream decision-making.

To ensure tractability in the downstream decision task, we model each uncertain parameter $y_i^{(k)}$ within $\mathbf{y}_i$ using a discrete distribution. The distribution of the $k$-th parameter is represented as

$$p_i(y^{(k)}) = \sum_{m=1}^{M} \pi_{i,m}^{(k)} \delta(y^{(k)} - \mu_{i,m}^{(k)}), \tag{2}$$

where $M$ is the number of mixture components (a hyperparameter), $\pi_{i,m}^{(k)} \geq 0$, $\sum_{m=1}^{M} \pi_{i,m}^{(k)} = 1$, and $\delta(\cdot)$ is the Dirac delta function. Each data point's support set is denoted as the vector $\boldsymbol{\mu}_i^{(k)} = (\mu_{i,1}^{(k)}, \ldots, \mu_{i,M}^{(k)})$, constructed individually based on approximation relationships rather than from predefined values.

In our framework, the predictive model outputs a distribution over uncertain parameters for each input, capturing multiple plausible outcomes. We then optimize the expected cost under this predicted distribution. In practice, we represent each marginal distribution as a finite mixture and solve a weighted empirical risk minimization over the mixture components. A detailed derivation and the full discrete-support formulation are provided in Appendix A.

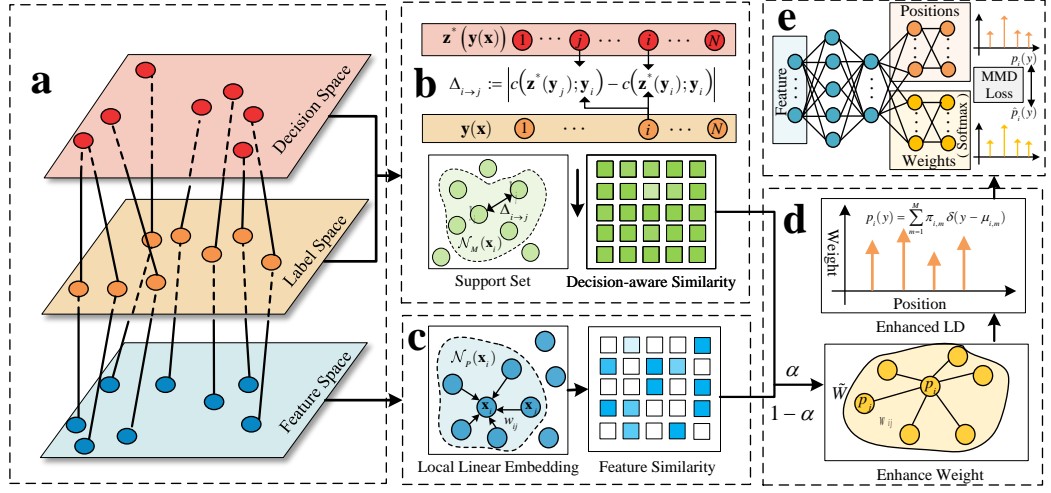

Figure 1: Overview of the Decision-aware LDL Framework; a) Mapping relationships; b) Mining decision information; c) Mining feature information; d) Constructing enhanced weights; e) Learning enhanced label distributions

## 4.2 LABEL ENHANCEMENT VIA LOCAL MANIFOLD AND TASK-DRIVEN GRAPH STRUCTURES

### 4.2.1 DECISION-AWARE LABEL SUPPORT CONSTRUCTION

A key insight of this paper is that, as shown in Figure 1(b), rather than redefining the loss function, we reconstruct the label manifold to embed decision-awareness into label representations. To this end, we define the optimization transfer cost difference from sample $j$ to $i$ as

$$\Delta_{j \to i} := \left| c(\mathbf{z}^*(\mathbf{y}_i), \mathbf{y}_i) - c(\mathbf{z}^*(\mathbf{y}_j), \mathbf{y}_i) \right|, \tag{3}$$

where $\mathbf{z}^*(\mathbf{y}_i)$ denotes the optimal decision under parameters $\mathbf{y}_i$, and $c(\mathbf{z}, \mathbf{y})$ is the task cost evaluated for decision $\mathbf{z}$ under parameters $\mathbf{y}$.

A smaller value of $\Delta_{j \to i}$ indicates that, in the decision problem associated with sample $i$, substituting $\mathbf{y}_j$ for the true parameters $\mathbf{y}_i$ incurs only a minor additional cost. Predictions with smaller $\Delta_{j \to i}$ are thus more acceptable from the perspective of downstream decision-making. We normalize this asymmetric transfer cost into a decision-aware similarity score. Since $\min_{r,l} \Delta_{r \to l} = 0$, we have

$$s_{j \to i} = 1 - \frac{\Delta_{j \to i}}{\max_{r,l=1,\dots,N} \Delta_{r \to l}}, \tag{4}$$

where higher values denote stronger optimization-level affinity.

Finally, we perform row-wise normalization so that the similarities sum to 1 for each target $i$:

$$\tilde{s}_{j \to i} = \frac{s_{j \to i}}{\sum_{r=1}^{N} s_{r \to i}}. \tag{5}$$

The resulting matrix $\tilde{S} = [\tilde{s}_{j \to i}]_{i,j=1}^{N} \in \mathbb{R}^{N \times N}$ encodes the decision-aware relational structure among samples, where its $(i, j)$-th entry $\tilde{s}_{j \to i}$ quantifies the transferability from sample $j$ to sample $i$ and is utilized in the label enhancement stage.

To convert the point-supervised target $\mathbf{y}_i$ into a mixture of Dirac delta functions, we construct an individualized support vector $\boldsymbol{\mu}_i^{(k)}$ for each $k$-th parameter of sample $i$. Unlike conventional approaches that define support points solely based on feature similarity, we propose to select them according to decision-aware similarity $\tilde{s}_{j \to i}$.

Specifically, we identify the top-$M$ neighbors whose decisions exhibit maximal transferability to sample $i$, characterized by the largest values of $\tilde{s}_{j \to i}$. Formally, the neighborhood is defined as

$$\mathcal{N}_M(\mathbf{x}_i) := \left\{ j \in \{1, \dots, n\} \mid \operatorname{rank}(\tilde{s}_{j \to i}) \leq M \right\},$$

where $\operatorname{rank}(\tilde{s}_{j \to i})$ denotes the rank of $\tilde{s}_{j \to i}$ in descending order among all values $\tilde{s}_{j \to i}$. The support vector corresponding to the $k$-th parameter is then defined as the ordered vector of the neighbor values:

$$\boldsymbol{\mu}_i^{(k)} = \left( y_j^{(k)} \right)_{j \in \mathcal{N}_M(\mathbf{x}_i)}. \tag{6}$$

This construction ensures that the support vector $\boldsymbol{\mu}_i^{(k)}$ for each sample $\mathbf{x}_i$ captures the local decision-level structure of the $M$ most transferable neighbors for the $k$-th parameter, thereby providing a decision-aware foundation for label distribution reconstruction. In this way, each label component is enriched to carry more information, and by selecting support values aligned with similar decisions, the support set further guides the predictive model toward outputs that induce lower decision errors.

### 4.2.2 DECISION-AWARE LABEL WEIGHTING VIA MANIFOLD RECONSTRUCTION

To assign weights to each $\mu_{i,m}^{(k)}$, as shown in Figure 1(c), we draw inspiration from manifold learning techniques that capture local geometric structures in the feature space. Formally, the feature-space neighborhood of a point $\mathbf{x}_i$ is defined as

$$\mathcal{N}_P(\mathbf{x}_i) := \left\{ j \in \{1, \dots, N\} \mid \operatorname{rank}\big(d(\mathbf{x}_i, \mathbf{x}_j)\big) \leq P, \ j \neq i \right\}, \tag{7}$$

where $d(\cdot, \cdot)$ denotes the distance metric in the feature space, $\operatorname{rank}\big(d(\mathbf{x}_i, \mathbf{x}_j)\big)$ is the ascending rank of the distance $d(\mathbf{x}_i, \mathbf{x}_j)$ among all other points with respect to $\mathbf{x}_i$, and $P$ is the number of nearest neighbors considered for each point (a hyperparameter).

Based on the neighborhood structure, we construct a local linear relationship by solving the following optimization problem. Let $W \in \mathbb{R}^{N \times N}$ denote the reconstruction weight matrix, whose $(i, j)$-th entry is $w_{ij}$. The weights are obtained by minimizing

$$\min_{W} \quad \Theta(W) := \sum_{i=1}^{N} \left\| \mathbf{x}_i - \sum_{j=1}^{N} w_{ij} \mathbf{x}_j \right\|^2, \tag{8}$$

subject to

$$\sum_{j=1}^{N} w_{ij} = 1, \quad w_{ij} = 0 \quad \text{if } j \notin \mathcal{N}_P(\mathbf{x}_i), \quad \forall i, j = 1, \dots, N. \tag{9}$$

The objective function in equation 8 seeks to represent each point $\mathbf{x}_i$ as a convex combination of its neighbors, minimizing the reconstruction error. The constraints in equation 9 restrict the reconstruction to the $P$-nearest neighbors, exclude self-reconstruction, and enforce convexity.

To further align label representation with the downstream decision task, we introduce a decision-aware correction directly in the optimization objective, rather than modifying the similarity matrix $W$ itself. Specifically, for the $k$-th uncertain parameter, we estimate the distribution weights $\boldsymbol{\pi}_i^{(k)} = (\pi_{i,1}^{(k)}, \dots, \pi_{i,M}^{(k)})$, representing the probability distribution over the $M$ values in the support vector $\boldsymbol{\mu}_i^{(k)}$. Let $u$ denote the index of the support value $\mu_{i,u}^{(k)}$ that corresponds to the ground-truth label of the $i$-th sample. The optimization problem is formulated as a convex combination of two consistency terms: one based on the feature-level similarity $W$ and the other on the task-induced similarity $\tilde{S}$:

$$
\begin{aligned}
\min_{\{\boldsymbol{\pi}_i^{(k)}\}} \quad \Psi(\boldsymbol{\pi}^{(k)}) := \sum_{i=1}^{N} \left\| \boldsymbol{\mu}_i^{(k)} \boldsymbol{\pi}_i^{(k)\top} - \sum_{j=1}^{N} w_{ij}\, \boldsymbol{\mu}_j^{(k)} \boldsymbol{\pi}_j^{(k)\top} \right\|^2 \\
+ \alpha \sum_{i=1}^{N} \left\| \boldsymbol{\mu}_i^{(k)} \boldsymbol{\pi}_i^{(k)\top} - \sum_{j=1}^{N} \tilde{s}_{ij}\, \boldsymbol{\mu}_j^{(k)} \boldsymbol{\pi}_j^{(k)\top} \right\|^2,
\end{aligned}
\tag{10}
$$

subject to

$$\sum_{m=1}^{M} \pi_{i,m}^{(k)} = 1, \qquad \forall i = 1, \dots, N, \tag{11}$$

$$\pi_{i,m}^{(k)} \geq 0, \qquad \forall i = 1, \dots, N, \ m = 1, \dots, M, \tag{12}$$

$$\pi_{i,u}^{(k)} \geq \lambda, \qquad \forall i = 1, \dots, N. \tag{13}$$

The objective in equation 10 enforces local consistency by matching the expected label values weighted by $\boldsymbol{\pi}_i$ with those of neighbors under both the feature-based similarity $W$ and the task-induced similarity $S$, combined through the trade-off parameter $\alpha$. The constraints in equation 11 ensure that each $\boldsymbol{\pi}_i$ forms a valid probability distribution by summing to one. The constraints in equation 12 guarantee non-negativity of all distribution components. Finally, the constraints in equation 13 enforce a minimum confidence $\lambda$ on the ground-truth label for each sample, thereby incorporating supervision into the manifold-based formulation.

### 4.3 Enhanced Label Distribution Learning with Neural Networks

As shown in Figure 1(e), given the enhanced dataset $\mathcal{D}' = \{(\mathbf{x}_i, p_i(\mathbf{y}))\}_{i=1}^{N}$ obtained via LE, we employ $K$ independent dual-branch neural networks $f_k(\cdot; \theta_k)$, $k = 1, \dots, K$, to predict the marginal distributions of the $K$ uncertain parameters individually, enabling the model to capture parameter-specific uncertainty as well as variations relevant to downstream decision-making.

For the $k$-th parameter, the network $f_k(\cdot; \theta_k)$ consists of an encoder and two specialized decoders for predicting mixture weights and support positions. The encoder maps the feature $\mathbf{x}_i$ through $L$ hidden layers to generate a parameter-specific representation

$$\mathbf{h}^{(L,k)} = f_{\text{enc}}^{(k)}(\mathbf{x}_i) \in \mathbb{R}^t,$$

where $t$ denotes the dimension of the encoder output, capturing the contextual information relevant to both decoder branches for the $k$-th parameter.

The decoders then compute the mixture weights and support positions as

$$\boldsymbol{\pi}^{(k)}(\mathbf{x}_i) = f_\pi^{(k)}\big(\mathbf{h}^{(L,k)}\big) \in \Delta^{M-1}, \tag{14}$$

$$\boldsymbol{\mu}^{(k)}(\mathbf{x}_i) = f_\mu^{(k)}\big(\mathbf{h}^{(L,k)}\big) \in \mathbb{R}^M, \tag{15}$$

where $f_\pi^{(k)}$ and $f_\mu^{(k)}$ denote the multi-layer decoders mapping the encoder output $\mathbf{h}^{(L,k)}$ to the respective mixture weights and support positions.

To measure the discrepancy between the predicted and target distributions for each parameter, we employ the Maximum Mean Discrepancy (MMD) metric, which quantifies the distance between distributions in a reproducing kernel Hilbert space (RKHS). The detailed derivation and closed-form expression of MMD for mixtures of Dirac delta functions are provided in Appendix B.

This design enables end-to-end learning of individualized label distributions for all $K$ parameters, preserving the geometric structure from the LE phase while aligning with decision-aware similarities—without requiring gradient flow through downstream optimization.

## 5 CASE STUDY

In this section, we evaluate the numerical performance of the proposed decision-aware LDL framework on both synthetic and real-world datasets. Synthetic data allow for controlled and reliable evaluation, while real-world data provide practical validation under realistic noise and annotation challenges. We exclude end-to-end learning frameworks typically designed for linear objectives or point estimates, as our nonlinear contextual stochastic optimization setting necessitates modeling the full conditional distribution. Consequently, our baseline selection adheres to established protocols for nonlinear objectives to ensure a methodologically aligned comparison. The following benchmark methods are included:

- **SAA**: This baseline disregards contextual features and determines decisions by minimizing the average cost under the empirical distribution of observed random parameters.

- **Prescriptive Analytics**: Following the framework of Bertsimas & Kallus (2020), we evaluate several local learning variants, including $k$-nearest neighbors (KNN), kernel regression (Kernel), local linear smoothing (LOESS), and classification and regression trees (CART tree).

- **Feature-based LDL**: As a strong baseline derived from our proposed method, this variant replaces the decision-aware similarity matrix $\mathbf{S}$ with a standard feature-based similarity. It can also be viewed as an ablation of our full framework, highlighting the contribution of decision-aware structure.

Details of the synthetic data generation process and the feature engineering procedures for both synthetic and real-world datasets (Buttler et al., 2022) are provided in the Appendix C. In our experiments, the synthetic data samples are drawn from a set of $n \in \{100, 200, 500, 700, 1000\}$, while the real-world datasets are constructed by rescaling historical data from years 1, 2, 3, and 4. To evaluate out-of-sample performance, we adopt distinct splitting strategies depending on the data type. For synthetic datasets, the data is randomly partitioned into training and test sets with an 80:20 ratio. In contrast, for real-world datasets, we strictly employ a chronological split to preserve temporal order: the first 80% of the historical data is used for training, while the subsequent 20% serves as the test set. We also present representative comparisons between feature similarity and decision similarity using heatmaps, as detailed in Section H.

### 5.1 MULTI-ITEM NEWSVENDOR PROBLEM

The multi-item Newsvendor problem seeks the optimal replenishment quantities for $K$ different products. Let $\mathbf{y} := (y_1, \dots, y_K)$ denote the random demand vector for the $K$ products, and let $\mathbf{z} \in \mathbb{R}^K$ represent the corresponding order quantities.

The demand $\mathbf{y}$ may depend on contextual factors such as promotions, holiday effects, or brand attributes. The total inventory cost consists of holding costs $h_k$ and stockout costs $b_k$, which penalize overstock and understock, respectively. Thus, the cost function is defined as:

$$c(\mathbf{z}, \mathbf{y}) := \sum_{k=1}^{K} h_k(z_k - y_k)^+ + b_k(y_k - z_k)^+,$$

where $(a)^+ := \max\{a, 0\}$ denotes the positive part function.

Additionally, we impose a budget constraint $C > 0$ on the total order quantities, leading to the following feasible set:

$$\mathcal{Z} := \left\{ \mathbf{z} \in \mathbb{R}^K : \sum_{k=1}^{K} z_k \leq C, \ \mathbf{z} \geq 0 \right\}.$$

We consider the case of $K = 2$, where the newsvendor jointly decides the order quantities for two products under a total budget constraint of 200. The unit overstock costs are set to $h_1 = 1$ and $h_2 = 1.3$, while the unit stockout costs are $b_1 = 9$ and $b_2 = 8$, respectively. For our decision-aware LDL model, the parameters are set as $P = M = 6$, $\alpha = 0.1$ and $\lambda = 0.3$.

Figure 2 and Figure 3 compare the test-set performance of decision-aware LDL and baseline approaches on synthetic and real-world data, respectively. Decision-aware LDL consistently achieves the lowest regret with strong stability, even in small-sample settings, demonstrating robustness across scenarios. Removing task-specific information increases both regret and variance. Overall, these results highlight that decision-aware LDL reliably improves decision quality in both controlled simulations and practical applications.

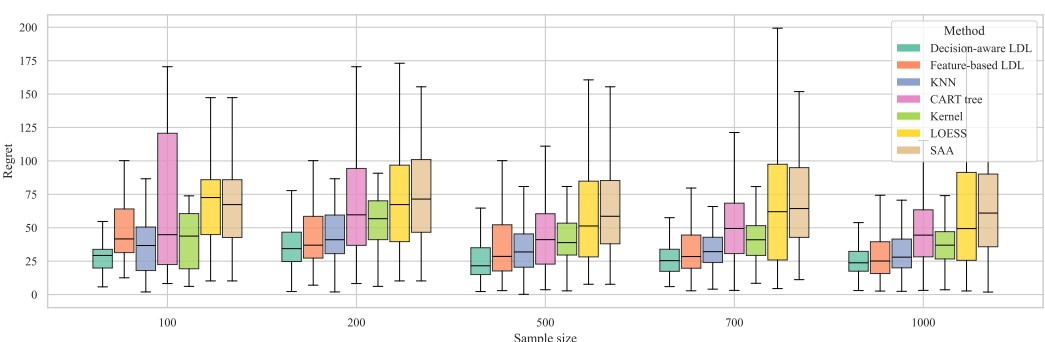

Figure 2: Comparison results for multi-item newsvendor problem in synthetic data.

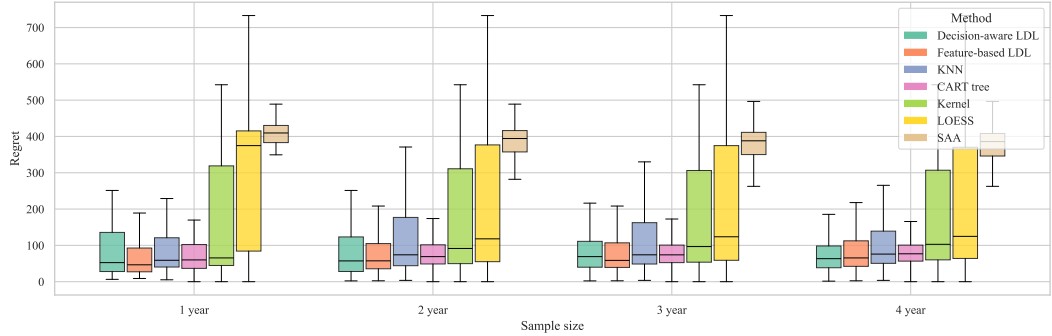

Figure 3: Comparison results for multi-item newsvendor problem in real-world data.

We further extend our evaluation to a discrete decision variant of the multi-item Newsvendor problem, where order quantities are restricted to specific values, rendering the objective function non-continuous. Detailed experimental results for this challenging setting are provided in Appendix D.

Consistent with the continuous cases, Decision-aware LDL outperforms all baselines, verifying its effectiveness even in rigid, non-smooth decision landscapes.

## 5.2 QUADRATIC COST NETWORK FLOW PROBLEM

Many applications such as urban traffic systems and communication networks can be formulated as a minimum convex cost flow problem. We consider a directed graph with $K$ edges, where the decision variable $\mathbf{z} = (z_1, \dots, z_K) \in \mathbb{R}^K$ denotes the flow on each edge, and $\mathbf{y} = (y_1, \dots, y_K) \in \mathbb{R}^K$ is a random parameter vector influencing the edge costs. The cost function is defined as

$$c(\mathbf{z}, \mathbf{y}) := \sum_{k=1}^{K} g_k(z_k, y_k), \tag{16}$$

where each $g_k(z_k, y_k)$ is a convex function of the flow $z_k$, and may vary across edges.

Let $A \in \mathbb{R}^{n \times K}$ be the node-arc incidence matrix of the graph, representing flow conservation at each node. In addition, let $C \in \mathbb{R}^{m \times K}$ be a constraint matrix that encodes edge- or path-based flow restrictions, with lower and upper bounds $\boldsymbol{\ell}, \mathbf{u} \in \mathbb{R}^m$. The feasible set is then expressed as

$$\mathcal{Z} := \left\{ \mathbf{z} \in \mathbb{R}^K : A\mathbf{z} = 0, \ \boldsymbol{\ell} \leq C\mathbf{z} \leq \mathbf{u} \right\}. \tag{17}$$

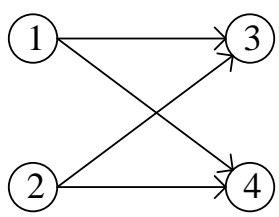

We consider a directed network with two source nodes (1 and 2) and two sink nodes (3 and 4), as illustrated in Figure 4. Let $z_1, z_2, z_3, z_4$ denote the flows on arcs (1,3), (1,4), (2,3), and (2,4), respectively. The flow on each arc incurs a convex cost of the form $g_k(z_k, y_k) = c_k(z_k - y_k)^2$, where $y_k$ is a random parameter and $c_1 = 1$, $c_2 = 3, c_3 = 2, c_4 = 2$ denote the cost coefficients for each arc. Each source node must send at least 10 units of flow, and each sink node must receive at least 10 units of flow.

Figure 4: Network Graph

For our decision-aware LDL model, we set $M = P = 6$, $\alpha = 0.1$ and $\lambda = 0.3$. Due to the symmetric quadratic objective, prediction errors have a relatively small effect on decision outcomes, so less emphasis is placed on decision-specific correlations.

Figure 5 shows that decision-aware LDL consistently achieves the lowest regret with strong stability. Removing decision-specific structure increases regret and variance, though the simplified version remains acceptable. As sample size grows, performance differences narrow, indicating that all methods approach optimal decisions with more information. Overall, across diverse problems, decision-aware LDL demonstrates robust and effective decision learning.

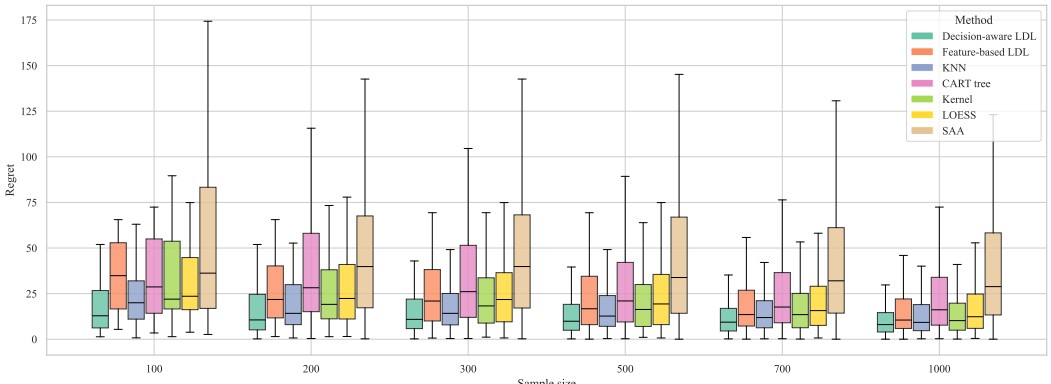

Figure 5: Comparison results for the minimum quadratic cost network flow problem.

## 5.3 COMPUTATIONAL EFFICIENCY

We provide a detailed breakdown of computational costs across training and inference phases in Appendix F (see Figures 7-9). The results reveal a clear strategic trade-off: while our decision-aware

LDL incurs a higher offline training cost due to the convex optimization required for solving the distribution weights $\boldsymbol{\pi}$, it offers significant advantages in online operations. Specifically, in complex optimization tasks like the Quadratic Cost Network Flow, the decision time for the SAA baseline scales linearly with the training sample size $N$, becoming computationally prohibitive for large datasets. In contrast, our method condenses historical information into a compact distribution estimate, ensuring that the online decision time remains constant and efficient regardless of the training size. This makes the proposed framework particularly well-suited for time-sensitive applications where offline training is permissible in exchange for rapid online decision-making.

### 5.4 Robustness and Sensitivity Analysis

To provide a comprehensive evaluation of model stability, we present the detailed numerical results—including the **mean and standard deviation** of out-of-sample costs and ranks—in Appendix E (see Tables 3–8). Across all three experimental settings (Synthetic Newsvendor, Real-world Newsvendor, and Quadratic Cost Network Flow), the proposed Decision-aware LDL consistently achieves the superior performance in terms of both average cost and ranking stability. Notably, in the real-world Newsvendor scenarios characterized by significant inherent noise, our method maintains low variance compared to the high volatility observed in traditional non-parametric baselines like LOESS and SAA. This empirical evidence confirms that the proposed manifold regularization effectively acts as a denoising filter, ensuring reliable decision-making even under distribution shifts and noisy supervision.

Regarding hyperparameter sensitivity, the extensive analysis provided in Appendix G (Figure 10) robustly demonstrates that the proposed method maintains low-regret performance across a wide range of configurations. We generally recommend setting the feature manifold neighbor count $P$ equal to the support size $M$ (typically $P = M = 6$) to align the geometric scope of the feature and decision spaces. Furthermore, introducing a small positive trade-off parameter (e.g., $\alpha \in [0.1, 0.5]$) is sufficient to incorporate the critical decision guidance. A notable phenomenon observed in this analysis is that as the training data size increases, the model's robustness improves, and concurrently, its sensitivity to parameter choices diminishes. Nevertheless, the overall finding is that the proposed methodology exhibits low sensitivity to hyperparameter settings.

## 6 Conclusion

Existing ILO approaches typically achieve decision-awareness by modifying the loss function of the predictor. However, the non-differentiable and discontinuous nature of decision-aware losses poses significant challenges for efficient training. In this work, we propose an alternative pathway that avoids loss modification.

Our decision-aware LDL framework provides a principled solution by modeling uncertainty as full distributions and strategically reallocating predictive mass away from high-risk regions. The approach transforms scalar targets into individualized mixture distributions using decision-aware similarity matrices, and employs a dual-branch neural network to learn decision-optimized representations. Experimental results on the newsvendor and network flow problems demonstrate consistent superiority in regret minimization across different sample sizes, with particularly strong performance in small-sample regimes where traditional methods struggle.

While promising, our approach has limitations including the conditional independence assumption for multivariate parameters and computational overhead of the label enhancement procedure. Future work should explore extensions to handle dependent parameters, develop efficient approximation techniques for large-scale applications, and provide theoretical performance guarantees. Nevertheless, this work establishes LDL as a viable paradigm for bridging statistical prediction and decision optimization, opening new research avenues at the intersection of machine learning and operations research.

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

# A DISCRETE MIXTURE REPRESENTATION FOR DECISION-AWARE LDL

Once the predictive function $f$ learns to output individualized mixture distributions $\mathcal{G}_i$ for each data point, we obtain, for each input $\mathbf{x}$, a collection of mixture parameters $\{(\pi_{k,m}(\mathbf{x}), \mu_{k,m}(\mathbf{x}))\}_{m=1}^{M}$ for each dimension $k = 1, \ldots, K$. Here, $K$ denotes the number of uncertain parameters (or output dimensions) involved in the decision problem.

In practice, we train $K$ separate predictive models, each dedicated to learning the mixture distribution of one uncertain parameter. That is, the $k$-th model outputs $\{(\pi_{k,m}(\mathbf{x}), \mu_{k,m}(\mathbf{x}))\}_{m=1}^{M}$, capturing the uncertainty associated with dimension $k$. This decomposition allows the framework to scale to high-dimensional decision problems while preserving interpretability at the marginal level.

Each marginal distribution is represented as a mixture of Dirac delta functions:

$$p_k(y_k \mid \mathbf{x}) = \sum_{m=1}^{M} \pi_{k,m}(\mathbf{x}) \, \delta(y_k - \mu_{k,m}(\mathbf{x})),$$

and under the conditional independence assumption, the joint distribution is

$$P(\mathbf{y} \mid \mathbf{x}) = \prod_{k=1}^{K} p_k(y_k \mid \mathbf{x}) = \sum_{\mathbf{m} \in [M]^K} \Pi_{\mathbf{m}}(\mathbf{x}) \, \delta(\mathbf{y} - \boldsymbol{\mu}_{\mathbf{m}}(\mathbf{x})),$$

where $\mathbf{m} = (m_1, \ldots, m_K)$ indexes one mixture component per dimension, $\boldsymbol{\mu}_{\mathbf{m}}(\mathbf{x}) = [\mu_{1,m_1}(\mathbf{x}), \ldots, \mu_{K,m_K}(\mathbf{x})]^T$, and $\Pi_{\mathbf{m}}(\mathbf{x}) = \prod_{k=1}^{K} \pi_{k,m_k}(\mathbf{x})$.

The expected cost under this distribution reduces to a weighted sum:

$$\mathbb{E}_{\mathbf{y} \sim P(\mathbf{y}|\mathbf{x})} [c(\mathbf{z}, \mathbf{y})] = \sum_{\mathbf{m} \in [M]^K} \Pi_{\mathbf{m}}(\mathbf{x}) \, c\left(\mathbf{z}, \boldsymbol{\mu}_{\mathbf{m}}(\mathbf{x})\right),$$

and the corresponding optimal decision is

$$\mathbf{z}_{\text{dist}}^*(\mathbf{x}) = \arg\min_{\mathbf{z} \in \mathcal{Z}} \sum_{\mathbf{m} \in [M]^K} \Pi_{\mathbf{m}}(\mathbf{x}) \cdot c\left(\mathbf{z}, \boldsymbol{\mu}_{\mathbf{m}}(\mathbf{x})\right).$$

# B MAXIMUM MEAN DISCREPANCY BETWEEN TWO MIXTURES OF DIRAC DELTA FUNCTIONS

Let us consider two probability distributions that are discrete mixtures of Dirac delta functions:

$$P = \sum_{i=1}^{m} a_i \, \delta(x - x_i), \qquad Q = \sum_{j=1}^{n} b_j \, \delta(y - y_j),$$

where each $x_i$ and $y_j$ is a point in the sample space $\mathcal{X}$, and the weights satisfy

$$a_i \geq 0, \quad b_j \geq 0, \quad \sum_{i=1}^{m} a_i = \sum_{j=1}^{n} b_j = 1.$$

Here, $\delta(\cdot)$ denotes the Dirac delta distribution, so that $\delta(x - x_i)$ places all of its probability mass at the point $x_i$.

Given a symmetric positive definite kernel function $k : \mathcal{X} \times \mathcal{X} \to \mathbb{R}$ (for example, the Gaussian radial basis function kernel), the squared *Maximum Mean Discrepancy* (MMD) between $P$ and $Q$ is defined as:

$$\text{MMD}^2(P, Q) = \mathbb{E}_{x,x' \sim P} \, k(x, x') + \mathbb{E}_{y,y' \sim Q} \, k(y, y') - 2 \, \mathbb{E}_{x \sim P, \, y \sim Q} \, k(x, y).$$

The MMD measures the distance between the *mean embeddings* of $P$ and $Q$ in the reproducing kernel Hilbert space (RKHS) induced by $k$. When $k$ is *characteristic*, $\text{MMD}^2(P, Q) = 0$ if and only if $P = Q$.

For discrete measures such as mixtures of Dirac deltas, the expectations above reduce to finite sums. Substituting the expressions for $P$ and $Q$ into the MMD definition yields:

$$\text{MMD}^2(P,Q) = \sum_{i=1}^{m} \sum_{i'=1}^{m} a_i a_{i'} \, k(x_i, x_{i'}) + \sum_{j=1}^{n} \sum_{j'=1}^{n} b_j b_{j'} \, k(y_j, y_{j'}) - 2 \sum_{i=1}^{m} \sum_{j=1}^{n} a_i b_j \, k(x_i, y_j).$$

This expression is exact and does not require any sampling, as all terms are directly computable from the given support points and weights.

It is often convenient to express the above in matrix notation. Define:

$$K_{XX}[i, i'] = k(x_i, x_{i'}), \quad K_{YY}[j, j'] = k(y_j, y_{j'}), \quad K_{XY}[i, j] = k(x_i, y_j),$$

and let $\mathbf{a} = (a_1, \ldots, a_m)^\top$, $\mathbf{b} = (b_1, \ldots, b_n)^\top$. Then:

$$\text{MMD}^2 = \mathbf{a}^\top K_{XX} \mathbf{a} + \mathbf{b}^\top K_{YY} \mathbf{b} - 2 \, \mathbf{a}^\top K_{XY} \mathbf{b}.$$

This compact form is particularly useful for implementation, since it involves only matrix–vector multiplications.

The above closed-form expression is valid for any positive definite kernel $k$. For the Gaussian RBF kernel:

$$k(x, y) = \exp\left( -\frac{\|x - y\|^2}{2\sigma^2} \right),$$

$\text{MMD}^2(P,Q)$ becomes a function of the pairwise squared Euclidean distances between $\{x_i\}$ and $\{y_j\}$, making it especially efficient to compute when these distances can be precomputed.

### SPECIAL CASE: MMD BETWEEN SINGLE-POINT DISTRIBUTIONS

We now analyze a special case to build intuition. Let us consider two distributions, $P$ and $Q$, where all probability mass is concentrated on a single point for each. That is, all support points for $P$ collapse to a single point $p$, and all support points for $Q$ collapse to a single point $q$:

$$x_1 = x_2 = \cdots = x_m = p \quad \text{and} \quad y_1 = y_2 = \cdots = y_n = q.$$

Effectively, this simplifies the distributions to $P = \delta(x - p)$ and $Q = \delta(y - q)$, as the weights must sum to one.

We substitute these conditions into the general MMD formula:

$$\begin{aligned}
\text{MMD}^2(P,Q) &= \sum_{i=1}^{m} \sum_{i'=1}^{m} a_i a_{i'} \, k(x_i, x_{i'}) + \sum_{j=1}^{n} \sum_{j'=1}^{n} b_j b_{j'} \, k(y_j, y_{j'}) - 2 \sum_{i=1}^{m} \sum_{j=1}^{n} a_i b_j \, k(x_i, y_j) \\
&= \sum_{i=1}^{m} \sum_{i'=1}^{m} a_i a_{i'} \, k(p, p) + \sum_{j=1}^{n} \sum_{j'=1}^{n} b_j b_{j'} \, k(q, q) - 2 \sum_{i=1}^{m} \sum_{j=1}^{n} a_i b_j \, k(p, q) \\
&= k(p, p) \left( \sum_{i=1}^{m} a_i \right) \left( \sum_{i'=1}^{m} a_{i'} \right) + k(q, q) \left( \sum_{j=1}^{n} b_j \right) \left( \sum_{j'=1}^{n} b_{j'} \right) \\
&\quad - 2 \, k(p, q) \left( \sum_{i=1}^{m} a_i \right) \left( \sum_{j=1}^{n} b_j \right) \\
&= k(p, p) \cdot (1) \cdot (1) + k(q, q) \cdot (1) \cdot (1) - 2 \, k(p, q) \cdot (1) \cdot (1) \\
&= k(p, p) + k(q, q) - 2 \, k(p, q).
\end{aligned}$$

This result is the squared distance between the embeddings of points $p$ and $q$ in the RKHS, i.e., $\text{MMD}^2(P,Q) = \|\phi(p) - \phi(q)\|_{\mathcal{H}}^2$, where $\phi(x) = k(\cdot, x)$ is the feature map.

**Example with Gaussian RBF Kernel** If we further substitute the Gaussian RBF kernel, $k(x, y) = \exp\left(-\frac{\|x-y\|^2}{2\sigma^2}\right)$, we note that $k(p, p) = \exp(0) = 1$ and $k(q, q) = \exp(0) = 1$. The expression simplifies to:

$$\text{MMD}^2(P, Q) = 1 + 1 - 2\exp\left(-\frac{\|p-q\|^2}{2\sigma^2}\right) = 2\left(1 - \exp\left(-\frac{\|p-q\|^2}{2\sigma^2}\right)\right).$$

This clearly shows that the MMD between two single-point distributions is a function of the Euclidean distance between the points $p$ and $q$. If $p = q$, the MMD is 0, as expected.

## C EXPERIMENTAL SETTING

### C.1 REAL-WORLD BAKERY DATA

For our experiments on real-world data, we use the bakery dataset from Buttler et al. (2022). We focus on two products from the same store. The target variable $y$ corresponds to product demand, while the feature set $X$ includes:

- Historical demand of the past week
- Holiday indicators: `is_schoolholiday`, `is_holiday`, `is_holiday_next2days`
- Weather-related features: `temp_min`, `temp_avg_celsius`, `temp_max`, `rain_mm`
- Promotion features: `promotion_currentweek`, `promotion_lastweek`
- Temporal features: `weekday`, `month`

All non-categorical features are normalized, while categorical features are encoded as one-hot vectors.

### C.2 SYNTHETIC DATA GENERATION

For synthetic experiments, we generate regression datasets using `make_regression`. Specifically:

- Newsvendor problem: 4 features, with $y$ scaled to the range [10, 120], and each demand's noise standard deviation $\sigma = 8$
- Quadratic cost network flow problem: 6 features, with $y$ scaled to the range [5, 15], and each arc's noise standard deviation $\sigma$ approximately 1.2

### C.3 NEURAL NETWORK CONFIGURATION

To jointly estimate the mixture weights and location parameters of the impulse signals, we implement a Multi-Task Learning (MTL) neural network. The architecture comprises a shared feature extraction backbone followed by two task-specific decoupling heads:

**Shared Encoder:** The backbone transforms the input $\mathbf{x}$ into a high-level latent representation via a series of fully connected blocks with decreasing dimensionality ($512 \to 256 \to 128$). Each block incorporates GELU activation, LayerNorm for stability, and Dropout ($p = 0.2$) to mitigate overfitting.

**Weight Prediction Head:** This Multi-Layer Perceptron (MLP) branch maps latent features to the mixture weights. A Softmax activation is applied to the final layer to strictly enforce the simplex constraint (i.e., $\sum \hat{w}_i = 1$).

**Position Prediction Head:** A parallel MLP branch designed to regress the continuous location parameters. It employs GELU activation in the hidden layers to capture non-linearities and outputs the raw coordinates via a linear projection.

**Optimization Setup.** The network is optimized using the Adam optimizer with an initial learning rate of $10^{-3}$ and a weight decay of $10^{-5}$. We employ a *ReduceLROnPlateau* scheduler (factor=0.5, patience=10) to anneal the learning rate when the validation loss plateaus. Furthermore, an early stopping mechanism with a patience of 20 epochs is utilized to ensure convergence and prevent overfitting.

### C.4 BASELINE IMPLEMENTATIONS AND HYPERPARAMETERS

To ensure a fair comparison and robust performance, we standardized the hyperparameter settings for the baseline methods as follows:

- **k-Nearest Neighbors (KNN):** We set the number of neighbors to $k = 10$. This value was chosen to strike a balance between local sensitivity (capturing the relevant neighborhood structure) and statistical stability (smoothing out observation noise).
- **CART (Decision Tree):** We restricted the tree complexity by setting `max_leaf_nodes=6` and enforcing `min_samples_leaf=10`. These constraints act as regularization to prevent the tree from overfitting to specific training samples, ensuring that each leaf node contains sufficient data to form a reliable empirical distribution.
- **Kernel Regression (Nadaraya-Watson):** We employed the Epanechnikov kernel with a baseline bandwidth of 1. Crucially, we enabled `use_variable_bandwidth=True`, which allows the estimator to adaptively adjust the smoothing window based on local data density, thereby improving performance in sparse regions of the feature space.
- **LOESS (Locally Estimated Scatterplot Smoothing):** We configured the local regression with an Epanechnikov kernel and set the neighborhood size to `n_neighbors=10`, consistent with the KNN baseline. Furthermore, we enabled `use_nonnegative=True` to enforce domain constraints, ensuring that the predicted demand or costs remain non-negative and physically feasible.
- **SAA (Sample Average Approximation):** As a feature-agnostic baseline, SAA assumes that the future distribution is identical to the unconditional empirical distribution of the training data. We implemented this by using the global pool of target samples, ignoring any conditional feature information $X$.

## D DISCRETE DECISION MULTI-ITEM NEWSVENDOR PROBLEM

In this section, we consider a variant of the multi-item Newsvendor problem where the order quantities are restricted to a discrete set of values (e.g., specific batch sizes or pallet counts). This introduces a non-convex, combinatorial aspect to the downstream optimization, serving as a stress test for the proposed method's capability to handle non-smooth decision landscapes.

### D.1 PROBLEM FORMULATION

Similar to the continuous case, let $\mathbf{y} := (y_1, \ldots, y_K)$ denote the random demand vector for $K$ products, and let $\mathbf{z} = (z_1, \ldots, z_K)$ represent the order quantities. The total inventory cost remains defined as the sum of holding and stockout costs:

$$c(\mathbf{z}, \mathbf{y}) := \sum_{k=1}^{K} h_k(z_k - y_k)^+ + b_k(y_k - z_k)^+, \tag{18}$$

where $(a)^+ := \max\{a, 0\}$. However, unlike the continuous setting, the decision variables are constrained to a discrete set of allowable order levels $\mathcal{Q} = \{q_1, q_2, \ldots, q_L\}$ where $L$ represents the number of candidate quantities. Consequently, the feasible set is defined as:

$$\mathcal{Z} := \left\{ \mathbf{z} \in \mathcal{Q}^K : \sum_{k=1}^{K} z_k \le C \right\}, \tag{19}$$

where $C$ is the total budget capacity. In our experiments, we set $\mathcal{Q} = \{10, 20, \ldots, 130\}$ and $C = 200$, forcing the optimization to select order quantities from fixed steps of 10 units.

## D.2 Experimental Results

. We evaluate the performance of Decision-aware LDL against baseline methods using synthetic datasets. The experimental setting aligns strictly with the synthetic data generation process described in the continuous Newsvendor case (Section 5), with sample sizes $N \in \{100, \ldots, 1000\}$. We restrict this analysis to synthetic scenarios because the significant volatility and irregular fluctuations inherent in real-world demand data make it practically infeasible to define a static, effective discrete decision set $\mathcal{Q}$ (e.g., fixed batch sizes) that remains valid across different temporal shifts.

The comparative results are summarized in Figure 6 and detailed in Tables 1 and 2.

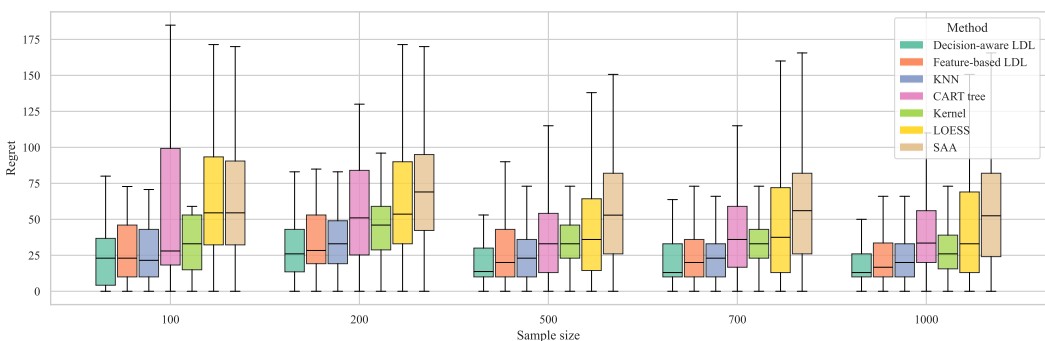

Figure 6: Comparison results for the discrete decision multi-item newsvendor problem. The proposed method maintains superiority even when decision variables are constrained to discrete levels.

Table 1: Out-of-sample costs (Mean ± Std) for the Discrete Newsvendor problem.

| Method | 100 | 200 | 500 | 700 | 1000 |
|---|---|---|---|---|---|
| **Decision-aware LDL** | **23.72 ± 21.13** | **30.81 ± 27.48** | **21.37 ± 22.27** | **23.11 ± 25.63** | **21.54 ± 23.31** |
| **Feature-based LDL** | 29.10 ± 25.02 | 36.09 ± 27.31 | 29.22 ± 29.97 | 28.11 ± 29.30 | 27.20 ± 31.90 |
| KNN | 28.27 ± 25.65 | 36.86 ± 26.50 | 26.00 ± 23.45 | 25.07 ± 22.26 | 23.20 ± 21.58 |
| CART tree | 107.98 ± 155.92 | 76.10 ± 96.08 | 47.58 ± 67.56 | 48.59 ± 59.38 | 46.29 ± 54.90 |
| Kernel | 32.58 ± 21.71 | 44.65 ± 21.97 | 33.78 ± 20.00 | 32.71 ± 19.24 | 30.11 ± 18.32 |
| LOESS | 74.60 ± 67.34 | 73.83 ± 66.21 | 52.20 ± 55.16 | 52.62 ± 52.58 | 50.49 ± 56.41 |
| SAA | 64.35 ± 45.25 | 71.68 ± 38.52 | 60.36 ± 47.84 | 61.50 ± 45.26 | 61.21 ± 53.06 |

Table 2: Out-of-sample ranks (Mean ± Std) for the Discrete Newsvendor problem.

| Method | 100 | 200 | 500 | 700 | 1000 |
|---|---|---|---|---|---|
| **Decision-aware LDL** | **3.05 ± 2.76** | **2.74 ± 3.06** | **2.78 ± 2.27** | **2.82 ± 2.60** | **2.87 ± 2.41** |
| **Feature-based LDL** | 3.40 ± 2.62 | 3.08 ± 2.90 | 3.27 ± 3.32 | 3.19 ± 2.90 | 3.15 ± 2.72 |
| **KNN** | **3.05 ± 2.21** | 3.05 ± 1.43 | 3.09 ± 1.68 | 3.09 ± 1.82 | 3.09 ± 1.78 |
| **CART tree** | 4.35 ± 5.69 | 4.54 ± 3.63 | 4.29 ± 3.50 | 4.44 ± 3.42 | 4.56 ± 3.40 |
| **Kernel** | 4.30 ± 2.77 | 4.34 ± 2.01 | 4.44 ± 2.46 | 4.34 ± 2.54 | 4.34 ± 2.45 |
| **LOESS** | 4.75 ± 3.28 | 4.65 ± 4.41 | 4.60 ± 4.16 | 4.56 ± 4.36 | 4.44 ± 4.41 |
| **SAA** | 5.10 ± 2.83 | 5.59 ± 2.62 | 5.53 ± 2.66 | 5.55 ± 2.52 | 5.54 ± 2.68 |

## E Detail of Numerical Experiment Results

We evaluate the proposed method against six baselines across three distinct settings: the synthetic Newsvendor problem, the real-world Newsvendor problem, and the Quadratic Cost Network Flow problem. To ensure a robust comparison and mitigate the skewing effect of extreme outliers in absolute costs, we further introduce a rank-based metric. Specifically, for each out-of-sample test instance, we rank all methods based on their realized costs. We then aggregate these instance-level

rankings to report the mean rank and its standard deviation. This approach allows us to assess the relative performance consistency of each method, avoiding conclusions that are disproportionately influenced by rare, high-cost scenarios.

### E.1 RESULTS ON SYNTHETIC NEWSVENDOR PROBLEM

Tables 3 and 4 summarize the out-of-sample costs and average ranks for the synthetic newsvendor problem. We define the average rank as $\text{Rank}_{\text{avg}}(M) = \frac{1}{N_{test}} \sum_{i=1}^{N_{test}} \text{rank}_i(M)$, where $N_{test}$ is the total number of test instances and $\text{rank}_i(M)$ denotes the rank of method $M$ on the $i$-th instance (with rank 1 indicating the lowest cost). The proposed Decision-aware LDL consistently outperforms the baselines, achieving the lowest costs across all training sample sizes ($N = 100$ to $1000$). Notably, the cost reduction is significant compared to the standard Feature-based LDL (e.g., 26.83 vs. 38.67 at $N = 1000$), verifying that incorporating task-induced similarity $\tilde{S}$ effectively guides the label distribution learning. While KNN performs competitively in terms of rank, our method maintains better stability as evidenced by the consistently lower rank scores (ranging from 2.40 to 2.63).

Table 3: Out-of-sample costs (Mean $\pm$ Std) for the synthetic Newsvendor problem under varying training sizes.

| Method | 100 | 200 | 500 | 700 | 1000 |
|---|---|---|---|---|---|
| **Decision-aware LDL** | **25.96 ± 14.73** | **42.19 ± 27.27** | **29.14 ± 22.68** | **28.45 ± 20.02** | **26.83 ± 18.56** |
| **Feature-based LDL** | 58.82 ± 49.37 | 52.80 ± 37.10 | 42.98 ± 40.36 | 41.39 ± 38.99 | 38.67 ± 38.98 |
| **KNN** | 38.34 ± 20.96 | 43.48 ± 24.53 | 34.23 ± 24.92 | 33.78 ± 23.47 | 31.94 ± 23.47 |
| **CART tree** | 119.26 ± 162.21 | 87.72 ± 100.21 | 56.95 ± 69.15 | 58.09 ± 60.71 | 55.72 ± 56.20 |
| **Kernel** | 40.48 ± 23.75 | 54.54 ± 22.94 | 43.23 ± 20.35 | 42.06 ± 18.16 | 39.43 ± 16.92 |
| **LOESS** | 84.84 ± 70.03 | 87.53 ± 75.49 | 68.92 ± 62.22 | 71.02 ± 61.54 | 68.77 ± 62.49 |
| **SAA** | 66.80 ± 37.24 | 75.44 ± 34.99 | 67.66 ± 44.17 | 70.01 ± 43.87 | 70.07 ± 51.35 |

Table 4: Average ranks (Mean $\pm$ Std) for the synthetic Newsvendor problem.

| Method | 100 | 200 | 500 | 700 | 1000 |
|---|---|---|---|---|---|
| **Decision-aware LDL** | **2.40 ± 2.17** | 2.92 ± 2.41 | **2.59 ± 1.89** | **2.60 ± 1.97** | **2.63 ± 2.04** |
| **Feature-based LDL** | 3.90 ± 3.36 | 3.12 ± 2.78 | 3.14 ± 3.46 | 3.08 ± 3.23 | 2.97 ± 3.08 |
| **KNN** | 3.30 ± 1.69 | **2.72 ± 1.75** | 2.92 ± 2.15 | 2.97 ± 2.13 | 2.98 ± 1.89 |
| **CART tree** | 4.30 ± 5.30 | 4.42 ± 4.02 | 4.43 ± 3.71 | 4.51 ± 3.68 | 4.56 ± 3.50 |
| **Kernel** | 3.95 ± 4.76 | 4.48 ± 2.83 | 4.64 ± 2.62 | 4.52 ± 2.60 | 4.56 ± 2.48 |
| **LOESS** | 5.18 ± 3.19 | 4.83 ± 4.77 | 4.83 ± 4.29 | 4.81 ± 4.52 | 4.76 ± 4.74 |
| **SAA** | 4.97 ± 2.80 | 5.50 ± 2.60 | 5.45 ± 2.49 | 5.52 ± 2.32 | 5.54 ± 2.50 |

### E.2 RESULTS ON REAL-WORLD DATA (NEWSVENDOR)

The results on real-world datasets, presented in Tables 5 and 6, highlight the robustness of our approach across varying scales of historical data (ranging from 1 to 4 years). As indicated by the large standard deviations (e.g., $\pm 169.56$ for the 1-year period), these datasets contain significant inherent noise and potential distribution drifts. Despite these challenges, Decision-aware LDL achieves the lowest cost when utilizing 2, 3, and 4 years of training data, and remains highly competitive with only 1 year of data. Traditional methods like SAA and LOESS suffer heavily from the data variability, yielding much higher costs. Crucially, the rank analysis shows that our method (ranks $2.58 - 3.00$) maintains superior reliability across different temporal horizons compared to CART or Kernel methods. This suggests that the manifold-based smoothing acts as an effective denoising filter, preventing overfitting to short-term noise while effectively leveraging longer historical records.

### E.3 RESULTS ON QUADRATIC COST NETWORK FLOW PROBLEM

In the more complex Quadratic Cost Network Flow problem (Tables 7 and 8), the advantage of the proposed method is most pronounced. Decision-aware LDL ranks first consistently across all

Table 5: Out-of-sample costs (Mean ± Std) for the real-world Newsvendor problem across four product groups.

| Method | 1 year | 2 year | 3 year | 4 year |
|---|---|---|---|---|
| **Decision-aware LDL** | 117.01 ± 169.56 | **101.05 ± 131.13** | **103.66 ± 126.17** | **97.05 ± 122.45** |
| **Feature-based LDL** | 170.18 ± 448.82 | 174.72 ± 422.46 | 143.67 ± 327.16 | 132.16 ± 284.89 |
| **KNN** | 133.94 ± 160.15 | 142.77 ± 148.83 | 138.86 ± 139.84 | 127.64 ± 125.99 |
| **CART tree** | **112.59 ± 213.88** | 129.55 ± 290.35 | 117.53 ± 222.74 | 111.93 ± 189.14 |
| **Kernel** | 174.47 ± 195.57 | 185.44 ± 254.24 | 176.33 ± 205.23 | 177.46 ± 184.96 |
| **LOESS** | 292.22 ± 175.08 | 213.08 ± 276.21 | 211.00 ± 229.62 | 209.10 ± 208.96 |
| **SAA** | 388.64 ± 107.71 | 375.35 ± 112.50 | 377.21 ± 129.86 | 375.67 ± 137.20 |

Table 6: Average ranks (Mean ± Std) for the real-world Newsvendor problem.

| Method | 1 year | 2 year | 3 year | 4 year |
|---|---|---|---|---|
| **Decision-aware LDL** | 2.86 ± 3.19 | **2.58 ± 3.16** | 3.00 ± 3.02 | **2.71 ± 2.99** |
| **Feature-based LDL** | **2.78 ± 3.16** | 3.07 ± 3.21 | **2.77 ± 3.35** | 2.86 ± 3.39 |
| **KNN** | 4.00 ± 3.15 | 4.02 ± 3.18 | 3.90 ± 3.31 | 3.82 ± 2.99 |
| **CART tree** | 3.40 ± 3.60 | 3.74 ± 3.42 | 3.81 ± 3.24 | 3.88 ± 2.98 |
| **Kernel** | 4.10 ± 2.19 | 4.31 ± 2.29 | 4.17 ± 2.48 | 4.30 ± 2.43 |
| **LOESS** | 4.96 ± 2.77 | 4.22 ± 3.22 | 4.25 ± 3.30 | 4.32 ± 3.12 |
| **SAA** | 5.90 ± 2.01 | 6.06 ± 1.99 | 6.09 ± 2.05 | 6.11 ± 2.21 |

sample sizes. As the training set size increases from $N = 100$ to 1000, the performance gap widens; for instance, at $N = 1000$, our method achieves a cost of 11.40 compared to 14.57 for the next best method (KNN) and 18.11 for Feature-based LDL. This confirms that for higher-dimensional, inter-dependent decision tasks, leveraging both feature and decision similarity is critical for learning accurate and decision-aligned parameters.

Table 7: Out-of-sample costs (Mean ± Std) for the Quadratic Cost Network Flow problem.

| Method | 100 | 200 | 300 | 500 | 700 | 1000 |
|---|---|---|---|---|---|---|
| **Decision-aware LDL** | **17.24 ± 14.06** | **17.39 ± 18.33** | **16.08 ± 15.32** | **14.06 ± 13.66** | **13.17 ± 12.75** | **11.40 ± 11.51** |
| **Feature-based LDL** | 35.71 ± 25.40 | 31.05 ± 29.91 | 29.40 ± 27.51 | 24.56 ± 25.06 | 21.31 ± 22.78 | 18.11 ± 20.83 |
| **KNN** | 22.25 ± 16.61 | 22.47 ± 22.71 | 21.05 ± 20.98 | 18.74 ± 18.74 | 16.96 ± 17.14 | 14.57 ± 15.42 |
| **CART tree** | 38.65 ± 37.26 | 42.11 ± 39.24 | 37.14 ± 37.94 | 32.11 ± 33.57 | 28.53 ± 30.81 | 26.40 ± 29.51 |
| **Kernel** | 36.17 ± 33.12 | 31.07 ± 31.75 | 27.33 ± 26.85 | 22.50 ± 22.59 | 19.30 ± 20.15 | 16.12 ± 17.75 |
| **LOESS** | 32.80 ± 24.26 | 33.32 ± 36.79 | 30.69 ± 31.94 | 27.36 ± 29.66 | 24.34 ± 28.04 | 20.96 ± 25.74 |
| **SAA** | 57.14 ± 51.47 | 56.84 ± 59.36 | 54.27 ± 53.30 | 49.72 ± 50.81 | 46.23 ± 49.06 | 44.06 ± 48.17 |

## F  COMPUTATIONAL EFFICIENCY ANALYSIS

Figures 7, 8, and 9 present a detailed breakdown of the computational time, categorized into **Training Time** and **Inference Time** (defined as the aggregate of the inference and decision phases across all out-of-sample datasets) for the Synthetic Newsvendor, Real-world Newsvendor, and Quadratic Cost Network Flow problems, respectively. To clearly visualize the varying magnitudes of computational cost, the top row of each figure displays the results on a linear scale, while the bottom row utilizes a logarithmic scale.

A consistent trend across all three figures is the strategic trade-off between offline training investments and online operational efficiency. Regarding the **Training Phase**, our proposed Decision-aware LDL incurs a higher computational cost compared to "lazy learning" methods like KNN (which effectively has zero training time) or SAA. This is attributable to the convex optimization required to learn the distribution weights $\pi$. However, this is strictly an offline cost that does not impede real-time applications.

The advantages of our method are most pronounced in the **Inference Phase** (online operation). Here, "Inference Time" captures the total latency from receiving a new feature $\mathbf{x}$ to generating the final de-

Table 8: Average ranks (Mean ± Std) for the Quadratic Cost Network Flow problem.

| Method | 100 | 200 | 300 | 500 | 700 | 1000 |
|---|---|---|---|---|---|---|
| **Decision-aware LDL** | **2.20 ± 2.59** | **2.25 ± 2.70** | **2.34 ± 2.70** | **2.40 ± 2.83** | **2.55 ± 2.91** | **2.60 ± 2.79** |
| **Feature-based LDL** | 4.10 ± 2.73 | 3.98 ± 3.37 | 4.13 ± 3.09 | 3.99 ± 3.27 | 3.85 ± 3.27 | 3.74 ± 3.42 |
| **KNN** | 3.25 ± 2.41 | 3.18 ± 2.12 | 3.09 ± 2.49 | 3.16 ± 2.66 | 3.19 ± 2.75 | 3.20 ± 2.78 |
| **CART tree** | 4.30 ± 6.54 | 4.62 ± 5.73 | 4.55 ± 5.46 | 4.46 ± 5.04 | 4.52 ± 4.80 | 4.64 ± 4.53 |
| **Kernel** | 4.20 ± 3.43 | 4.00 ± 2.58 | 3.81 ± 2.17 | 3.75 ± 2.03 | 3.49 ± 2.12 | 3.43 ± 1.95 |
| **LOESS** | 4.10 ± 1.46 | 4.20 ± 1.42 | 4.07 ± 1.76 | 4.26 ± 1.79 | 4.28 ± 1.84 | 4.25 ± 2.05 |
| **SAA** | 5.85 ± 2.56 | 5.77 ± 3.16 | 6.01 ± 2.53 | 5.98 ± 2.95 | 6.12 ± 2.48 | 6.14 ± 2.46 |

cision $\mathbf{z}^*$. This duration is heavily influenced by the number of support points involved. While some traditional methods incorporate all historical data into the decision-making process—significantly impairing computational efficiency—our method requires predicting only a **finite** set of support points, thereby achieving a **superior** inference time.

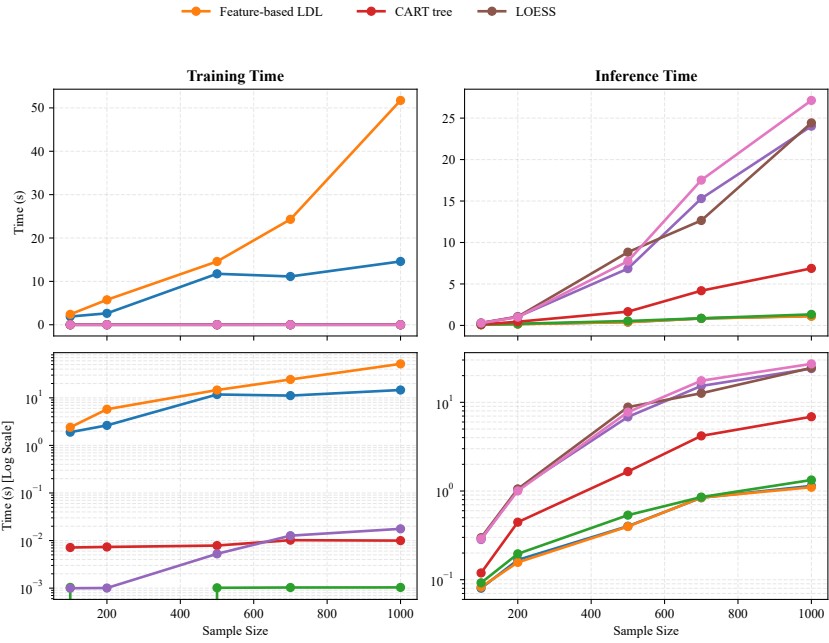

Figure 7: Computational efficiency breakdown for the **Synthetic Newsvendor Problem**.

# G SENSITIVITY ANALYSIS

## G.1 HYPERPARAMETER DISCUSSION

Our framework relies on four hyperparameters: support size $M$, manifold neighbors $P$, trade-off $\alpha$, and confidence threshold $\lambda$. We briefly discuss their roles and recommended settings:

- **Support Size ($M$) and Manifold Neighbors ($P$):** $M$ determines the granularity of the label distribution, while $P$ controls the locality of feature reconstruction. *Recommendation:* To align the geometric scope of the feature manifold with the decision support, we generally set $\mathbf{P} = \mathbf{M}$. A value in the range $[4, 10]$ is typically effective; excessively small values lead to sparse distributions, while larger values introduce noise from distant neighbors. In our experiments, we set $M = P = 6$.

- **Trade-off Parameter ($\alpha$):** This scalar controls the weight of the decision-aware consistency term relative to the feature-based term. *Recommendation:* A small positive value

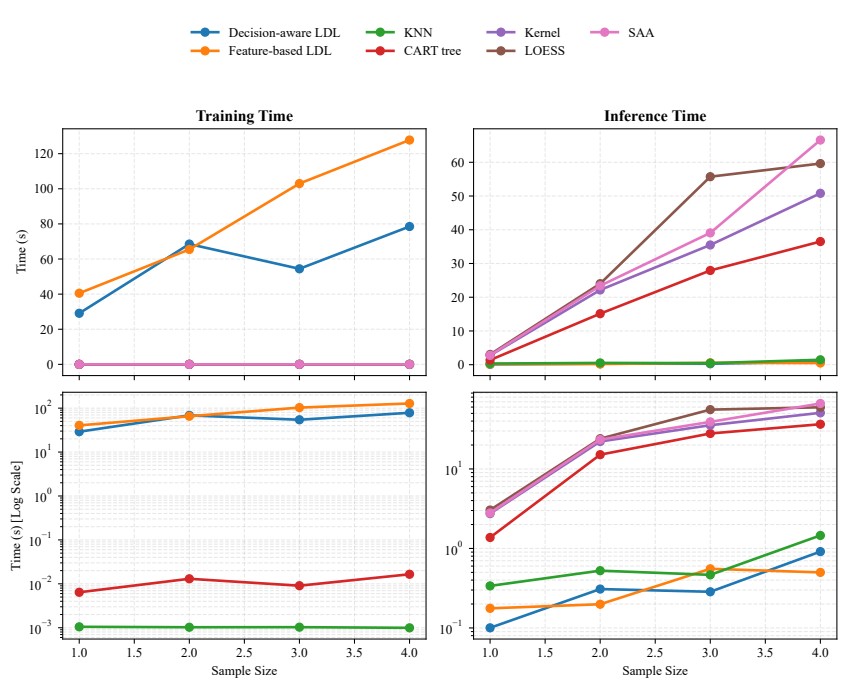

Figure 8: Computational efficiency breakdown for the **Real-world Newsvendor Problem** across historical data spans (1 to 4 years).

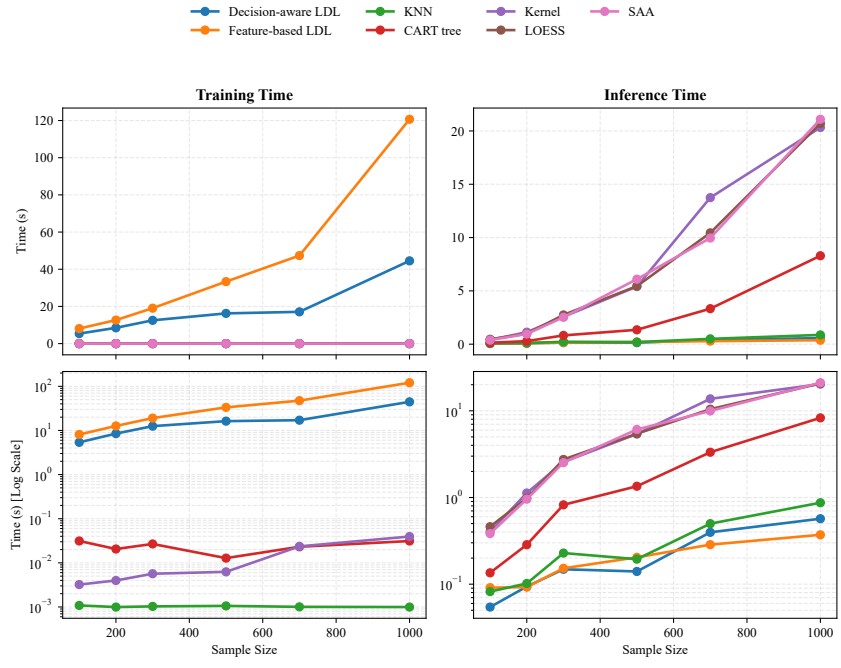

Figure 9: Computational efficiency breakdown for the **Quadratic Cost Network Flow Problem**.

(e.g., $\alpha \in [0.1, 0.5]$) is sufficient to introduce decision guidance without overriding the intrinsic feature semantics. We recommend $\alpha = 0.1$ as a robust baseline.

- **Confidence Threshold** ($\lambda$)**:** This parameter sets the minimum probability mass anchored to the ground-truth label. *Recommendation:* A moderate threshold (e.g., $\lambda \in [0.3, 0.5]$) is preferred. It prevents the distribution from collapsing into a single Dirac delta function (preserving uncertainty) while ensuring the enhanced label does not drift too far from the observed supervision.

### G.2 EXPERIMENTAL ANALYSIS

To empirically evaluate robustness, we focus on the sensitivity of $M$ and $\alpha$, as summarized in Figure 10.

**Impact of Support Size $M$.** The top row of Figure 10 depicts average regret as $M$ varies from 4 to 14. In the Synthetic Newsvendor problem (top-left), regret increases with excessively large $M$ in small samples ($N = 100$), as the support set disperses probability mass to irrelevant neighbors. Conversely, the Quadratic Cost Network Flow problem (top-right) exhibits exceptionally flat curves. This stability is attributed to the **symmetric nature of the objective function**, which renders the downstream decision less sensitive to minor variations in support structure compared to the asymmetric Newsvendor cost.

**Impact of Trade-off Parameter $\alpha$.** The bottom row of Figure 10 examines $\alpha \in [0, 1]$. Consistent with the analysis of $M$, the Network Flow problem shows minimal sensitivity. In the Synthetic Newsvendor problem, particularly for small samples, we observe a slight convexity minimizing regret around $\alpha \in [0.2, 0.6]$. This confirms that combining both feature and task consistency yields better generalization when data is scarce, while moderate $\alpha$ values prevent the decision manifold from dominating feature signals.

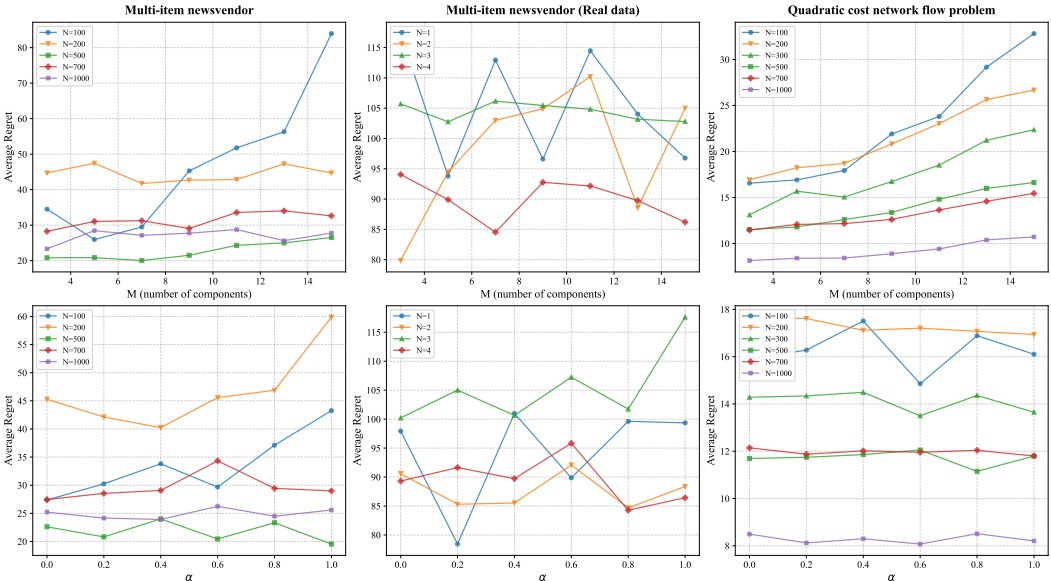

Figure 10: Sensitivity analysis of the Decision-aware LDL method regarding support size $M$ (top) and trade-off $\alpha$ (bottom).

## H DECISION AND FEATURE SIMILARITY ANALYSIS

Figures 11, 12, and 13 present representative similarity matrices derived from the minimum training dataset size. Our analysis draws a direct contrast between the feature similarity—quantified through a standard proximity measure (e.g., Gaussian distance) in the input feature space—and the decision

similarity, which corresponds to the matrix $\tilde{S}$ defined in the main text (see equation 3–equation 5). The principal conclusion emanating from this comparative visualization is that a fundamental misalignment exists between the structural similarity of the input features and the functional similarity of the corresponding optimal decisions.

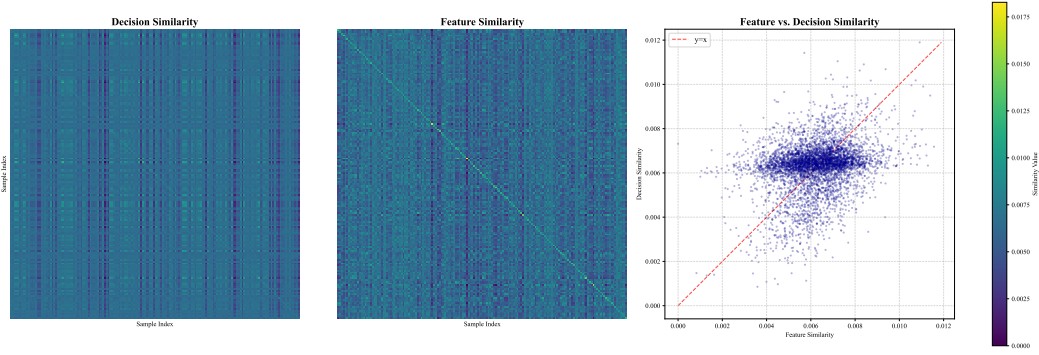

Figure 11: Newsvendor Problem Similarity Matrix

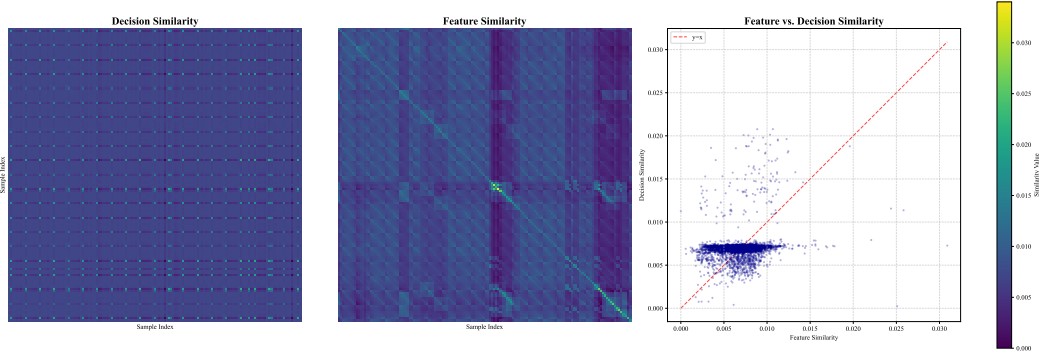

Figure 12: Real-World Newsvendor Problem Similarity Matrix

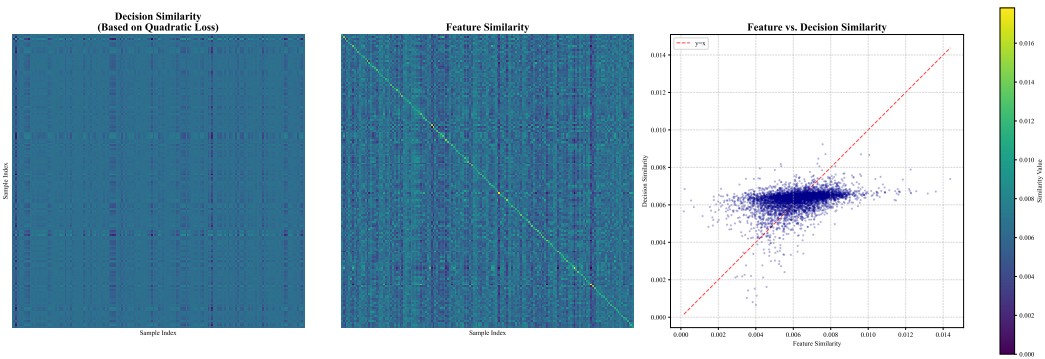

Figure 13: Quadratic Cost Network Flow Problem Similarity Matrix

## I    A COMPARISON EXAMPLE BETWEEN FIXED AND ADAPTIVE SUPPORT SET

In this section, we analyze the inherent limitations of using fixed support points. Let us revisit the newsvendor optimization problem described in Section 5. Conventional approaches for predicting

discrete distributions typically rely on discretizing the continuous demand space. For example, given a demand domain of $[0, 1000]$, the space is partitioned into equidistant intervals (or bins), and the model predicts the probability mass assigned to each bin.

This fixed discretization strategy suffers from two critical issues. First, the pre-defined support range acts as a hard constraint; if the actual demand falls outside this range, the model fails to capture the distribution tail. Second, this approach faces a fundamental **trade-off between precision and computational efficiency**. To minimize quantization errors, one must increase the resolution of the grid (i.e., use finer bins), but this drastically expands the dimensionality of the output space and increases computational complexity. Conversely, using coarser intervals to maintain efficiency (e.g., treating the range $[0, 10]$ as a single block) inevitably results in a loss of distributional information.

In contrast, our proposed method eliminates the dependency on fixed partitioning. By utilizing an adaptive support set, our model can directly predict precise support points and their corresponding probabilities—for instance, assigning a probability of $0.6$ to a demand of $100$ and $0.4$ to a demand of $120$—thus preserving the fidelity of the distribution without incurring the cost of high-dimensional discretization.

## J    MULTI-ITEM NEWSVENDOR PROBLEM WITH 10 ITEMS

To further evaluate the scalability and robustness of our proposed method in higher-dimensional decision spaces, we extended the experimental setting to a multi-item Newsvendor problem involving 10 distinct items. In this specific configuration, the unit holding costs $h$ were set to be linearly spaced in the interval $[1, 2]$, and the unit backorder costs $b$ were linearly spaced in the interval $[10, 20]$. Additionally, a shared capacity constraint of $C = 1000$ was imposed across all items. Apart from these specific adjustments to accommodate the increased dimensionality, all other experimental parameters remained consistent with those described in the main text.

The comparative results, visualized in Figure 14 and detailed in Tables 9 and 10, demonstrate that Decision-aware LDL consistently yields the lowest out-of-sample costs across most training sizes ($N \in \{100, 200, 500, 700\}$). Specifically, in data-scarce scenarios ($N = 100$), our method significantly outperforms the Feature-based LDL and KNN baselines. Although KNN shows competitive performance as the sample size increases to $N = 1000$, our method maintains a robust performance profile. These findings align with the conclusions drawn in the main text, confirming that Decision-aware LDL effectively captures the underlying decision landscape and yields superior performance even in higher-dimensional settings.

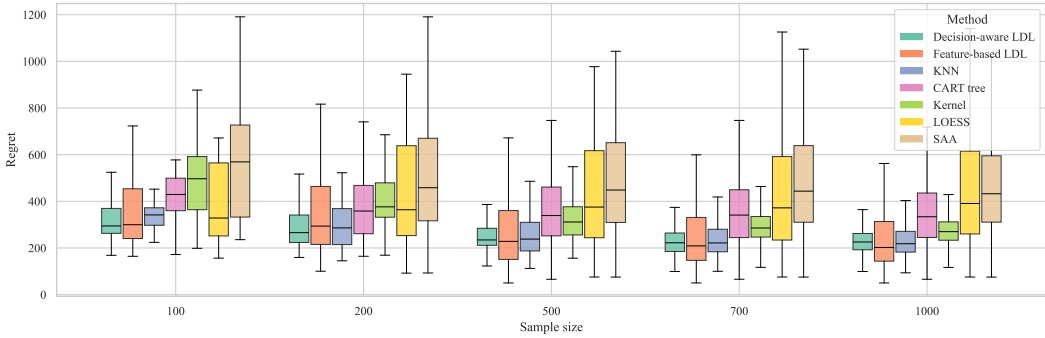

Figure 14: Comparison results for the multi-item newsvendor problem with 10 items.

## K    USE OF LARGE LANGUAGE MODELS IN MANUSCRIPT PREPARATION

During the preparation of this manuscript, large language models (LLMs) were occasionally employed to assist with tasks such as improving grammar, refining wording, and drafting certain sections of the text. These tools were used as aids to enhance clarity and readability, while all scientific content, analyses, results, and interpretations were developed and verified solely by the authors.

Table 9: Out-of-sample costs (Mean $\pm$ Std) for the Newsvendor problem with 10 items under varying training sizes.

| Method | 100 | 200 | 500 | 700 | 1000 |
|---|---|---|---|---|---|
| Decision-aware LDL | **327.58 $\pm$ 106.27** | **302.92 $\pm$ 140.32** | **260.85 $\pm$ 103.82** | **239.58 $\pm$ 98.39** | 238.91 $\pm$ 86.04 |
| Feature-based LDL | 376.43 $\pm$ 199.40 | 381.37 $\pm$ 308.12 | 298.32 $\pm$ 248.56 | 281.90 $\pm$ 247.43 | 268.66 $\pm$ 223.87 |
| KNN | 355.41 $\pm$ 129.63 | 310.47 $\pm$ 124.79 | 264.83 $\pm$ 110.64 | 245.32 $\pm$ 94.17 | **235.52 $\pm$ 83.90** |
| CART tree | 439.79 $\pm$ 136.06 | 430.95 $\pm$ 320.19 | 391.58 $\pm$ 241.85 | 398.93 $\pm$ 289.65 | 384.19 $\pm$ 260.42 |
| Kernel | 488.12 $\pm$ 164.43 | 405.98 $\pm$ 131.86 | 333.78 $\pm$ 112.39 | 303.82 $\pm$ 97.66 | 281.78 $\pm$ 88.11 |
| LOESS | 474.59 $\pm$ 421.73 | 520.73 $\pm$ 417.78 | 486.26 $\pm$ 382.87 | 483.93 $\pm$ 394.68 | 488.54 $\pm$ 362.68 |
| SAA | 569.50 $\pm$ 256.87 | 537.74 $\pm$ 323.75 | 516.30 $\pm$ 306.22 | 525.97 $\pm$ 347.93 | 499.55 $\pm$ 318.05 |

Table 10: Out-of-sample ranks (Mean $\pm$ Std) for the Newsvendor problem with 10 items under varying training sizes.

| Method | 100 | 200 | 500 | 700 | 1000 |
|---|---|---|---|---|---|
| **Decision-aware LDL** | **2.70 $\pm$ 1.17** | **2.75 $\pm$ 1.21** | **2.77 $\pm$ 1.24** | 2.69 $\pm$ 1.18 | 2.83 $\pm$ 1.35 |
| **Feature-based LDL** | 2.85 $\pm$ 4.98 | 3.25 $\pm$ 5.14 | 2.68 $\pm$ 4.03 | **2.63 $\pm$ 3.82** | **2.65 $\pm$ 3.87** |
| **KNN** | 3.30 $\pm$ 1.48 | 3.06 $\pm$ 1.88 | 2.84 $\pm$ 1.61 | 2.82 $\pm$ 1.45 | 2.70 $\pm$ 1.48 |
| **CART tree** | 4.65 $\pm$ 2.45 | 4.32 $\pm$ 3.20 | 4.59 $\pm$ 3.06 | 4.75 $\pm$ 3.11 | 4.74 $\pm$ 2.92 |
| **Kernel** | 5.20 $\pm$ 1.85 | 4.81 $\pm$ 2.48 | 4.48 $\pm$ 2.18 | 4.34 $\pm$ 2.12 | 4.03 $\pm$ 2.21 |
| **LOESS** | 3.65 $\pm$ 5.61 | 4.47 $\pm$ 5.42 | 4.88 $\pm$ 4.44 | 4.94 $\pm$ 4.31 | 5.22 $\pm$ 3.81 |
| **SAA** | 5.65 $\pm$ 3.19 | 5.35 $\pm$ 3.03 | 5.76 $\pm$ 2.35 | 5.83 $\pm$ 2.03 | 5.83 $\pm$ 1.96 |

The use of LLMs did not influence the originality of the research, the formulation of hypotheses, the design of experiments, or the interpretation of results. The authors have carefully reviewed and edited all content generated with the assistance of LLMs to ensure accuracy, consistency, and adherence to the manuscript's scientific standards.

