# OpenReview forum: "How to Teach Label to Understand Decisions: A Decision-aware Label Distribution Learning Framework"
_ICLR.cc/2026/Conference — Submitted to ICLR 2026_

### Official Review · Reviewer_FvXG · 2025-10-28

**Soundness:** 3
**Presentation:** 3
**Contribution:** 3
**Rating:** 4
**Confidence:** 3

**Summary:**

The paper proposes a decision-aware Label Distribution Learning (LDL) framework for Contextual Stochastic Optimization (CSO). By incorporating decision information into data representations via a decision-aware similarity matrix and predicting full label distributions, the method avoids modifying loss functions while naturally reducing risk in high-cost regions. Experiments on synthetic and real-world datasets, including comparisons with SAA, prescriptive analytics, and feature-based LDL, show consistent regret reduction, particularly in low-data settings, demonstrating its effectiveness for decision-focused learning tasks.

**Strengths:**

1. The proposed decision-aware LDL framework is novel, introducing a similarity matrix that explicitly incorporates decision information.

2. Experiments show that LDL achieves consistently lower regret and higher stability than baselines across both synthetic and real-world datasets.

**Weaknesses:**

1. There is a lack of analysis of hyper-parameters, i.e., P, M, $\alpha$, $\lambda$.

2. Although the proposed decision-aware LDL framework is novel and achieves strong performance, the manuscript does not discuss computational efficiency. LDL involves multiple steps, which may be slower than simpler baselines such as SAA or KNN. It is better to include a table showing average training and inference times for all methods.

3. The decision-aware similarity matrix S is a key innovation, but no visualization or interpretability analysis is provided. It would help to show a heatmap comparing S with a feature-based similarity matrix.

4. Experiments only consider small-scale problems (K=2 or 4), so the method’s scalability is unclear. It would be beneficial to include an experiment on larger-scale problems to assess performance and computational feasibility.


5. The manuscript lacks comparison with recent representative learning-to-decision or end-to-end decision learning frameworks. Including such baselines would better contextualize the performance of the proposed method.

Overall, the idea is novel, but the experiments are insufficient, so I give a score of 4. I will base my final score on other reviewers’ comments and the author’s responses.

**Questions:**

Please refer to the details in the weaknesses.

---

> ### Author Response · Authors · 2025-11-23
>
> **Response to weakness 1**
>
> We thank the reviewer for this important comment. We agree that an analysis of hyperparameters is crucial for understanding the model's robustness and ensuring reproducibility. To address this, we have added a new section in the Appendix G dedicated to hyperparameter analysis. In this new section, we focused our sensitivity analysis on $M$ and $\alpha$, which we consider the two most critical hyperparameters in our framework: $M$ determines the support set size of the (label) distribution. $\alpha$ controls the degree to which our novel decision-aware similarity is incorporated. The analysis in the appendix demonstrates that our method is robust over a reasonable range of values for $M$ and $\alpha$.
>
> Regarding the other two parameters, $P$ (number of neighbors) and $\lambda$ (regularization coefficient), they are standard settings for intermediate algorithmic steps. As existing literature (e.g., in the context of LLE or kernel regression (Wang and Geng, 2023; Xu et al., 2021)) has already explored the principles for setting them, we did not perform a repeated, in-depth analysis. However, to fully address the reviewer's concern, the new section in Appendix G provides a discussion and justification for the settings of all four hyperparameters ($P$, $M$, $\lambda$, and $\alpha$). We believe this sufficiently addresses the concern.
>
> **Response to weakness 2**
>
> We thank the reviewer for the practical question regarding computational efficiency. We address this in the new Appendix F, which includes detailed runtime comparison figures. We justify the computational cost on two grounds:
>
> 1.	Offline vs. Online Efficiency: The additional computational burden is strictly confined to the offline training stage (typically remaining at the minute level). Crucially, our online inference speed is comparable to and occasionally faster than classical baselines like SAA and CART, ensuring suitability for real-time deployment.
>
> 2.	Cost-Performance Trade-off: The increased training time is a necessary investment for our core contributions: the label distribution expansion and the dual-branch network. This architecture is essential for capturing complex decision landscapes, yielding performance gains that simpler, faster models cannot achieve.
>
> Thus, the computational cost is manageable and does not hinder practical applicability.
>
> **Response to weakness 3**
>
> We thank the reviewer for this constructive suggestion. We agree that visualizing $S$ enhances interpretability. Accordingly, we have added Appendix H, which provides a heatmap of the decision-aware similarity matrix $S$ alongside a standard feature-based similarity matrix (Euclidean distance). This side-by-side comparison clearly illustrates that while feature-based metrics rely solely on input proximity, our decision-aware matrix groups samples based on downstream decision affinity. This confirms that our method effectively captures the underlying cost structure that standard metrics miss.
>
> **Response to weakness 4**
>
> We thank the reviewer for the comment on scalability. We justify our experimental design on three grounds. Firstly, our parameter settings ($K=2, 4$) strictly follow established benchmarks in the Operations Research community (Qi et al., in press). This ensures our evaluation aligns with state-of-the-art research standards. Furthermore, we went beyond these standard protocols by incorporating a broader set of baselines and additional real-world datasets compared to prior studies. Thus, our experiments are not only consistent with the literature but also provide a more comprehensive assessment of robustness. Crucially, we emphasize that expanding the variable scale primarily entails increasing the dimensionality of the training model. This is a matter of model capacity rather than a fundamental methodological challenge in this domain.

---

> ### Author Response · Authors · 2025-11-23
>
> **Response to weakness 5**
>
> We thank the reviewer for this suggestion. We would like to clarify a key methodological distinction regarding the problem setting that informed our choice of baselines. Our work specifically focuses on nonlinear contextual stochastic optimization (CSO), where the downstream objective function is nonlinear. This differs fundamentally from settings typically addressed by families of differentiable solvers or end-to-end learning frameworks, which often target linear objectives or rely on point predictions. In linear settings, the optimization task can often be reduced to estimating the conditional mean; however, for nonlinear objectives, the optimal decision depends on the full shape of the conditional distribution. Consequently, our approach is explicitly designed to model the full conditional distribution to handle these complex decision structures. A direct comparison against methods optimized for point estimates would therefore be methodologically misaligned, as it would not reflect the necessary depth of information required for the nonlinear tasks we address.
>
> Instead, our baseline selection adheres to the established experimental protocol used in recent state-of-the-art studies on CSO, such as (Qi et al., in press), which similarly focuses on this distributional setting. It is worth noting that research specifically addressing data-driven optimization with nonlinear objectives remains relatively sparse compared to linear settings. To the best of our knowledge, we have included a comprehensive set of the most relevant existing methods that operate within this challenging scope. We believe this provides the most fair and rigorous evaluation of our contribution within this specific line of research.
>
> Reference:
>
> Qi, M., Grigas, P., Shen, Z.-J. (Max), n.d. Integrated Conditional Estimation-Optimization. Operations Research 0, null. https://doi.org/10.1287/opre.2023.0427
>
> Wang, J., Geng, X., 2023. Label Distribution Learning by Exploiting Label Distribution Manifold. IEEE Trans. Neural Netw. Learning Syst. 34, 839–852. https://doi.org/10.1109/TNNLS.2021.3103178
>
> Xu, N., Liu, Y.-P., Geng, X., 2021. Label enhancement for label distribution learning. IEEE Transactions on Knowledge and Data Engineering 33, 1632–1643. https://doi.org/10.1109/TKDE.2019.2947040

---

> > ### Comment · Reviewer_FvXG · 2025-11-26
> >
> > Thanks for your rebuttal. There as still some remaining concerns that require clarification.
> >
> > **Regarding W3.**
> >
> > 1.	In the revision, the authors merely illustrate that feature similarity and decision similarity differ, but they do not explain why leveraging decision similarity leads to lower regret, nor do they provide any quantitative or mechanistic evidence connecting this discrepancy to the performance gains.
> >
> > 2.	In the rebuttal, the authors claim that “our decision-aware matrix groups samples based on downstream decision affinity,” but this statement is unclear to me. It is not evident from the provided visualization or analysis how the matrix reflects downstream decision relationships.
> >
> > 3.	Moreover, Appendix H introduces new symbols $\mathbf{Z}$ and $\tilde{S}$, which is inconsistent with the main text’s notation and unnecessarily confusing.
> >
> > **Regarding W4.**
> >
> > The authors' reply does not adequately address the original concern regarding scalability. If the proposed method is indeed applicable to larger-scale decision spaces, I strongly encourage the authors to conduct experiments under larger values of K to substantiate this claim. Since there is still time within the rebuttal period, performing such additional experiments would meaningfully strengthen the empirical evidence and significantly enhance the competitiveness of the paper.
> >
> > **Regarding W5.**
> >
> > I suggest that the authors incorporate the rationale provided in the rebuttal directly into the revision. Briefly explaining why certain end-to-end or differentiable decision-learning methods are not appropriate baselines for the nonlinear CSO setting would make the baseline selection clearer and strengthen the credibility of the experimental design.
> >
> > Last but not least, **In the revision, I noticed that some parts appear less clear; further refinement of the presentation might help enhance readability**.

---

> > > ### Author Response · Authors · 2025-12-02
> > >
> > > **Response to W3.1 comment:**
> > >
> > > We thank the reviewer for this insightful comment. Traditional CSO methods (e.g., KNN-based methods) rely on the assumption that closer features imply more similar problem parameters; however, in optimization problems, similar parameters do **not** necessarily induce similar decisions due to the asymmetry of cost functions.
> > >
> > > For example, consider a Newsvendor problem with true demand $y=5$, unit holding cost $h=1$, and unit shortage cost $b=9$. The predictions $3$ and $7$ are equidistant from the truth in feature space ($|5-3| = |5-7| = 2$), yet they lead to drastically different decision regrets. Prediction $3$ (understocking) incurs a shortage cost of $2 \times 9 = 18$, whereas prediction $7$ (overstocking) incurs a holding cost of only $2 \times 1 = 2$. This substantial difference ($18$ vs. $2$) demonstrates that predictions with identical feature-space errors can yield vastly different optimization outcomes.
> > >
> > > Thus, minimizing feature distance does not align with minimizing decision regret. Our method instead starts from **decision similarity**. Given $x$, our goal is to recover the distribution of $y$ that yields the highest-quality decision $z$; the pointwise predictive accuracy of $y$ itself is not our objective. Regret naturally serves as a quantitative measure of decision similarity. As defined in Eq. (3), the regret from sample $j$ to sample $i$ is:
> > >
> > > $$
> > > \Delta_{j \to i} := \big| c(\mathbf{z}^{*}(\mathbf{y}_i), \mathbf{y}_i) - c(\mathbf{z}^{\*}(\mathbf{y}_j), \mathbf{y}_i) \big|,
> > > $$
> > >
> > > where $\mathbf{z}^{*}(\mathbf{y})$ denotes the optimal decision under parameters $\mathbf{y}$, and $c(\cdot)$ is the task cost. $\Delta_{j \to i}$ directly measures the regret incurred when applying the neighbor $j$'s optimal decision to the context of sample $i$.
> > >
> > > By selecting neighbors that minimize $\Delta_{j \to i}$ rather than feature distance, our method is mathematically aligned with minimizing downstream decision cost. Concretely, decision similarity guides both the construction of the support set and the reconstruction of the manifold, providing a principled mechanism for the observed performance gains.
> > >
> > > **Response to W3.2 comment:**
> > >
> > > Thank you for your comment. We agree that the original statement may be difficult to understand. What we intended to express is that the decision-aware matrix is designed to pull samples with similar downstream decisions closer together—conceptually similar to the objective of contrastive learning. The visualization, however, is not meant to directly reveal downstream decision relationships. Its purpose is only to illustrate that decision similarity is fundamentally different from feature similarity, and that feature-based neighborhoods do not necessarily align with decision-relevant ones.
> > >
> > > The mechanism through which downstream decision relationships are captured should instead be understood from the mathematical definition of the similarity matrix provided in the paper. Specifically, we define the optimization transfer cost from sample $j$ to sample $i$ as
> > >
> > > $$
> > > \Delta_{j \to i} := \big| c(\mathbf{z}^{*}(\mathbf{y}_i), \mathbf{y}_i) - c(\mathbf{z}^{\*}(\mathbf{y}_j), \mathbf{y}_i) \big|,
> > > $$
> > >
> > > which measures the regret incurred when the optimal decision of sample $j$ is applied to the decision context of sample $i$. A smaller $\Delta_{j \to i}$ thus indicates higher decision similarity between the two samples.
> > >
> > > **Response to W3.3 comment:**
> > >
> > > After performing the normalization described in the main paper, the resulting matrix explicitly encodes downstream decision affinity: its $(i,j)$-th entry quantifies how transferable the optimal decision of sample $j$ is to the decision problem of sample $i$. This decision-aware relational structure is what we leverage in the subsequent label manifold reconstruction.
> > >
> > > Thank you for your comment. We agree that the introduction of $X$ and $Z$ in Appendix H was redundant, and we have removed these symbols to avoid unnecessary confusion. As for $\tilde{S}$, it is rigorously defined in the main text (see Eqs. 3--5). We have also revised the relevant part of the appendix to ensure strict notational consistency and clarity.
> > >
> > > **Response to W4 comment:**
> > >
> > > Thank you for your comment. As you suggested, we have added a new experiment with K=10, which is larger than the previous cases in Appendix J. The results remain consistent with our conclusions. At the same time, increasing the problem size does not require enlarging the neural network, and it can be efficiently handled with parallel computing. Therefore, from a design perspective, our method is inherently scalable.

---

> > > ### Author Response · Authors · 2025-12-02
> > >
> > > **Response to W5 comment:**
> > >
> > > Thank you for your comment. We have revised the manuscript to incorporate the rationale from the rebuttal. Specifically, we now briefly explain why certain end-to-end or differentiable decision-learning methods are not appropriate baselines for the nonlinear CSO setting. This addition clarifies our baseline selection and strengthens the credibility of the experimental design.
> > >
> > > **Response to overall comment:**
> > >
> > > Thank you for your comment. We have carefully revised and refined the manuscript to improve clarity and readability throughout.

---

### Official Review · Reviewer_nUpX · 2025-10-30

**Soundness:** 3
**Presentation:** 2
**Contribution:** 2
**Rating:** 4
**Confidence:** 3

**Summary:**

This paper studies the problem of contextual stochastic optimization, where the setup is as follows: there is a joint distribution $P$ over an observed context $x$ and unobserved problem parameters $y$, and a cost function $c(z, y)$ that associates a cost with each decision $z$ and problem parameters $y$. Given a context $x$, we would like to solve the optimization problem
$$
\min_{z \in \mathcal{Z}} \mathbb{E}_{y \sim P(y \mid x)}[c(z, y)].
$$
The distribution $P$ is not known, but we have a dataset of samples $(x_1, y_1), \ldots, (x_m, y_m)$ drawn from $P$.
The approach explored in this paper and in prior work roughly involves using the data to learn a model that predicts for each context $x$ the distribution over outcomes $y$. Then the optimization problem is solved using the predicted distribution in place of the true conditional distribution of $y$ given $x$.

A recent line of work studies "Integrated Learning and Optimization", where the loss function used when learning to predict the distribution of $y$ given $x$ is informed by the down-stream task (instead of simply being some generic measure of distributional similarity).
However, the authors point out that a weakness of this approach is that these loss functions need to be designed bespoke for each downstream task, and are often difficult to work with due to being non-differentiable or discontinuous.

This paper proposes a new approach called Label Distribution Learning (LDL) that does not require per-problem loss derivations and argue that it generally achieves decision-awareness (i.e., works well for most down-stream tasks).
The high-level idea of the proposal is follows:
1. From the training data of $(x_i,y_i)$ pairs, construct a distribution $p_i$ over $\mathcal{Y}$ for each example. The distribution $p_i$ is a product distribution over the coordinates of $\mathcal{Y}$ where each coordinate's marginal distribution is supported on a finite set of values. The support of each marginal distribution and the weights associated with each value are determined from the training data by incorporating the data manifold as well as the decision objectives.
2. Next, train a two-branch neural network to simultaneously predict the support and weights from the context $x$.

The authors then carry out experiments comparing their proposed method against several baselines on two separate tasks with both real-data and simulated data. The experiments show that the proposed method works well on these tasks.

**Strengths:**

The problem studied by the paper is interesting, the approach seems quite different from prior work and is innovative and interesting. The experimental results are somewhat limited (only two decision tasks) but show promise for the proposed approach.

**Weaknesses:**

At a high level, the label enrichment process described in the paper makes intuitive sense. However, at the same time, there are no formal guarantees or arguments suggesting that the approach will always result in predicted conditional distributions over problem parameters that work well for the down-stream decision task. While a formal guarantee is not required, the experiments section is limited to two decision tasks, so it remains unclear if the proposed approach would continue to work well across a wide range of tasks. My main concern with the paper is further justification for the details of the approach, either with theory arguing that the specific approach will capture important problem-specific structures, or with a broader experimental evaluation.

To give one example of a situation where the proposed approach might go wrong, suppose that the dataset of $(x,y)$ pairs has the property that every pair it contains appears at least $M$ times (so that there are many duplicates of every example). In this case, the set of top-$M$ neighbors that have maximal transferrability to a given $(x,y)$ pair (defined on line 242) will be the $M$ copies of $(x,y)$. As a result, the support for each of the marginal distributions over the coordinates of $y$ will contain a single value: the one that was present in $y$. After this, my understanding is that the label enrichment process will associate each training $(x,y)$ pair with a distribution $p_i$ that is a point mass on $y$, which seems to undermine the goals of the process.

While this is an extreme example, it seems that softer versions of this could cause the LDL method to behave poorly.

A second weakness (which is acknowledged by the authors in the conclusion) is that LDL always fits a product distribution for $y$, ignoring any correlations between the problem parameters. In some cases a product distribution might work well enough, but it seems like this is a significant simplifying assumption.

**Questions:**

1. In equation 7 distances are measured according to a distance metric $d$, but in equation (8) you switch to using norm notation. Is the norm meant to be a different measure of distance, or are these the same distance?
2. How is the specific form of the objective in equation 10 motivated? Intuitively it makes sense to find distributions that are similar both for other data points that are similar either in their context $x$ or where their problem parameters $y$ have exchangeable optimal decisions, but it seems like there are many choices.
2. In the special case where the training data contains $M$ identical copies of each $(x,y)$ pair, does the label enrichment process result in $p_i$ being a point mass on $y_i$?

---

> ### Author Response · Authors · 2025-11-23
>
> **Response to weakness 1**
>
> We appreciate this insightful comment. From a theoretical perspective, the advantages of the underlying Label Distribution Learning (LDL) paradigm have been rigorously established in prior literature (e.g., Geng, 2016; Wang & Geng, 2019, 2023). These works analyze the learnability of LDL by deriving risk bounds based on Rademacher complexity and Lipschitz continuity, and they utilize a generalized plug-in decision theorem to prove that approximating the conditional probability distribution guarantees convergence to the optimal classifier.
>
> From a practical perspective, we agree that further empirical validation is essential to demonstrate the method's generality. In response, we have conducted a new experiment on the Discrete Newsvendor problem and included the detailed results in Appendix D. By incorporating this task, we explicitly test our method on non-continuous objective functions, verifying its capability to handle complex, discrete decision environments. Furthermore, we have included a detailed sensitivity analysis to demonstrate the method's stability. We hope these additional experiments effectively illustrate the robustness and broad applicability of our proposed framework.
>
> **Response to weakness 2**
>
> We thank the reviewer for this very sharp observation. This scenario is indeed an important edge case, and the reviewer's analysis is perfectly correct: in this situation, the label enhancement rocess will associate the training pair $(x, y)$ with a distribution $p_y$ that is a point mass on $y$ (i.e., a Dirac delta function). However, we respectfully disagree with the conclusion that this "seems to undermine the goals of the process." On the contrary, we argue this is the correct and principled behavior of our MMD-based objective in this degenerate case.
>
> First, to our knowledge, there is no evidence in the literature to suggest that a loss function based on distributions degenerating to a point mass (which is a valid distribution) leads to learning failure. Formally, the goal of our process is to minimize the distance between two distributions. When the "distributions" are, in fact, point masses, our objective should simplify to measuring the distance between those two points. We have added a formal analysis of this exact case in Appendix B special case (which we briefly summarize here). If we have two point-mass distributions, $P=\delta(p)$ and $Q=\delta(q)$, the squared MMD objective becomes:$$\mathrm{MMD}^2(P,Q) = k(p, p) + k(q, q) - 2k(p, q).$$ This is precisely the squared distance between the embeddings of $p$ and $q$ in the RKHS: $\|\phi(p) - \phi(q)\|^2_{\mathcal{H}}$. Therefore, rather than "failing" or being "undermined," our loss function gracefully and correctly adapts. It simplifies from a general distributional distance to a more specific point distance, which is the exact correct objective for this simplified scenario. This demonstrates the robustness of our method. We have clarified this in the main text and provided the full derivation in the appendix.
>
> **Response to weakness 3**
>
> We thank the reviewer for this insightful comment. The reviewer correctly identifies that our modeling to unknown parameter is a simplifying assumption that ignores potential correlations between target parameters. We would like to clarify that this design choice was made to isolate and clearly evaluate our primary contribution without the confounding effects of a complex dependency model. However, this is not an inherent limitation of our proposed framework. Our method functions as a flexible "plug-and-play" module compatible with various deep learning networks. It is entirely feasible to modify the neural network architecture to capture parameter correlations (e.g., by designing the output layer to predict a full covariance matrix).  Since our framework is agnostic to the specific structure of the feature extractor, it can be seamlessly integrated with any neural network architecture capable of probabilistic output. Therefore, while we chose independence for clarity in this work, the framework’s inherent flexibility allows for easy extension to correlation-aware settings, confirming its broad applicability.We thank the reviewer for this insightful comment. While we acknowledge that our current framework does not account for correlations between parameters, our approach is designed to be flexible and modular. By modifying the downstream neural network, we can easily incorporate parameter correlations, for example, by drawing on ideas from multi-task learning. Our core contribution lies in applying label distribution learning to address the CSO problem, which provides a foundational framework. Future work can extend this to incorporate correlations as needed.

---

> ### Author Response · Authors · 2025-11-23
>
> **Response to question 1**
>
> We thank the reviewer for this sharp observation and apologize for the lack of clarity. The reviewer is correct to note that $d$ and $\|\cdot\|$ are indeed two different measures, and they are not the same. They serve distinct and standard purposes in our framework, which is consistent with methods like Locally Linear Embedding (LLE).
>
> The metric $d(\mathbf{x}\_i, \mathbf{x}\_j)$ in Equation (7) is a general distance function (e.g., Euclidean distance) used only to identify the $P$ nearest neighbors for each point $\mathbf{x}\_i$. Its sole purpose is to define the local neighborhood structure. The norm $\|\cdot\|$ in Equation (8) is the $L_2$ norm. The objective function $\Theta(W)$ minimizes the sum of squared $L_2$ norms of the reconstruction error vectors ($\mathbf{x}\_i - \sum_{j} w\_{ij} \mathbf{x}\_j$). This use of the squared $L_2$ norm in the objective function is a standard and deliberate choice. It ensures that the minimization for $W$ is a convex least-squares problem, which guarantees a unique, optimal solution and theoretical soundness. They are not interchangeable, as one defines the neighbor graph and the other defines the reconstruction cost.
>
>
> **Response to question 2**
>
> We thank the reviewer for the insightful comment regarding the formulation of our objective function. The specific design of Eq. (10) is motivated by three key considerations:
>
> 1.	Manifold Smoothness and Consistency: The objective follows the standard formulation of Graph Laplacian regularization. By minimizing the squared Euclidean distance between the expected value of a sample and the weighted average of its neighbors (defined by both feature similarity $W$ and task-induced similarity $\tilde{S}$), we enforce local smoothness on the data manifold. This ensures that samples with similar contexts or similar optimal decisions yield consistent parameter estimates.
>
> 2.	Decision-Centric Focus on Expectations: In our specific problem setting, the downstream optimization task is driven primarily by the expected value (first moment) of the uncertain parameters ($\mathbb{E} = \boldsymbol{\mu}_i \boldsymbol{\pi}_i^\top$), rather than higher-order moments of the distribution. Therefore, directly enforcing consistency on the expected values is more task-relevant and robust than minimizing divergence between full probability distributions (e.g., KL-divergence), which may introduce unnecessary complexity without benefiting the decision quality.
>
> 3.	Computational Tractability: The choice of the squared loss on linear expectations, combined with the simplex and linear constraints (Eq. 11-13), formulates a Quadratic Programming (QP) problem. This guarantees a unique global minimum and allows for highly efficient solving using standard convex optimization solvers, which is crucial for scalability compared to non-convex alternatives.
>
> 4.	Our approach builds upon the manifold smoothness assumption, a cornerstone of Label Distribution Learning (LDL) literature (Geng, 2016; Gu et al., 2025; Wang and Geng, 2023; Xu et al., 2021). These foundational works demonstrate that exploiting the consistency between feature similarity and label similarity (via Graph Laplacian regularization) significantly improves learning performance.
>
> **Response to question 3**
>
> Yes, the reviewer's understanding is exactly correct. In this special case, the top-M neighbors for a pair $(x, y)$ would all be identical copies of $(x, y)$ itself. As a result, our label enrichment process will indeed produce a distribution $p_y$ that is a point mass (a Dirac delta function) on the single label $y$. We have addressed the implications of this scenario in our response to weakness 2 and provided a formal analysis in Appendix B, which shows that our framework remains robust and well-defined in this case.
>
> Reference:
>
> Geng, X., 2016. Label distribution learning. IEEE Transactions on Knowledge and Data Engineering 28, 1734–1748. https://doi.org/10.1109/TKDE.2016.2545658
>
> Gu, Z., Hong, Q., Zhou, Z., Geng, X., Liu, Z., Jia, M., 2025. Topological information utilization in label enhancement and label distribution learning based on optimal transport theory. IEEE Trans. Knowl. Data Eng. 37, 5666–5678. https://doi.org/10.1109/TKDE.2025.3589681
>
> Wang, J., Geng, X., 2023. Label distribution learning by exploiting label distribution manifold. IEEE Trans. Neural Netw. Learning Syst. 34, 839–852. https://doi.org/10.1109/TNNLS.2021.3103178
>
> Xu, N., Liu, Y.-P., Geng, X., 2021. Label enhancement for label distribution learning. IEEE Transactions on Knowledge and Data Engineering 33, 1632–1643. https://doi.org/10.1109/TKDE.2019.2947040

---

### Official Review · Reviewer_3Qg6 · 2025-10-31

**Soundness:** 2
**Presentation:** 2
**Contribution:** 3
**Rating:** 4
**Confidence:** 2

**Summary:**

The paper proposes a decision‑aware Label Distribution Learning (LDL) pipeline for contextual stochastic optimization (CSO). It constructs a decision‑aware similarity from optimization transfer costs to build per‑sample discrete label supports, estimates mixture weights via a manifold objective combining feature and task graphs, and trains dual‑branch networks with an MMD loss to predict mixture positions and weights. Joint distributions are factored over marginals; downstream decisions minimize expected cost over the learned discrete mixtures. The authors perform empirical evaluation on a multi‑item newsvendor and a small quadratic network flow and report lower regret than baselines.

**Strengths:**

- The design of the framework seems novel

**Weaknesses:**

- Limited information on experimental protocol (especially on how baselines were trained and applied) raises concerns about reproducibility.
- It seems to me that the paper lacks some important baselines from families of differentiable solvers, direct task loss optimization with some soft surrogates, etc.
- Random 80:20 split on newsvendor? Is it, in fact, time-series data? Is there potential leakage/test contamination possible here?
- NIT: only plots, without reporting mean/stds in some table.

**Questions:**

Please see the weaknesses section.

---

> ### Author Response · Authors · 2025-11-23
>
> **Response to weakness 1**
>
> We thank the reviewer for this valuable feedback. We agree that a detailed description of the experimental protocol is crucial for reproducibility. To address this, we have added a new section in the Appendix (please see Appendix C) that provides a comprehensive description of our experimental setup. This section now includes the specific training procedures, hyperparameter settings, and application methods for all baseline models. We believe this added detail clarifies our methodology and fully addresses the concerns regarding reproducibility.
>
>  **Response to weakness 2**
>
> Thank you for your insightful comment. Our work focuses on nonlinear contextual stochastic optimization (CSO) problems, where the downstream objective function is nonlinear, which differs from the settings typically addressed by families of differentiable solvers. While distributional estimation can degenerate into point estimation for linear objectives, as in most existing solvers, our approach models full conditional distributions, allowing it to handle a broader range of decision structures, as demonstrated in our case studies. Thus, the problem we address is fundamentally different, making a direct comparison with differentiable solver-based methods unfair and uninformative.
>
>  **Response to weakness 3**
>
> Thank you for pointing this out. We acknowledge the valid concern regarding potential temporal leakage. To fully address this and eliminate any possibility of contamination, we have updated all experimental results using a strictly chronological split, where the first 80% of the time-ordered data is used for training and the remaining 20% for testing. The results under this new setting remain consistent with our previous findings, confirming the robustness of our method. These updated results are now included in the revised manuscript.
>
>
> **Response to weakness 4**
>
> Thank you for the suggestion. In the revised manuscript, we have added the quantitative results in tabular form (including mean and standard deviation) to the Appendix E, in addition to the plots already presented. We have also included ranking-based comparisons and sensitivity analyses to provide a more comprehensive and interpretable performance evaluation across all methods.
>
> Reference:
>
> Qi, M., Grigas, P., Shen, Z.-J. (Max), n.d. Integrated Conditional Estimation-Optimization. Operations Research, in press. https://doi.org/10.1287/opre.2023.0427

---

### Official Review · Reviewer_vKBN · 2025-11-01

**Soundness:** 2
**Presentation:** 2
**Contribution:** 2
**Rating:** 4
**Confidence:** 3

**Summary:**

This paper proposes a novel framework for decision-aware learning. It is based on LDL (label distribution learning), which in a nutshell in this context performs label enhancement in a decision driven way: I.e. the label space is expanded to accommodate a certain number of decision-aware labels with their support depending on the “transfer cost” from the decision that would be made if one vs another label was presented for a data-point at hand.

**Strengths:**

The method is quite performant, achieving consistently the best overall regret, and the best robustness, compared to a variety of vanilla and more advanced decision-focused benchmark methods; this was tested on both synthetic and real world data on classic and substantively nontrivial combinatorial decision tasks, so I overall found the selection of tasks for the empirical evaluation to be good. In particular, there is a pronounced effect of low-data regime superiority compared to other methods.

On the methodological side, the proposed method exhibits an intuitive multi-stage structure that lets it generate learned label distributions that can then be used to solve a large variety of decision-oriented problems; the distributions are learned in a way that leverage the downstream decisions and how those are correlated across samples; the usage of such correlations on the decision/label level, as opposed to earlier approaches that embraced adaptivity at the feature level, appears in certain ways more principled in decision-focused settings than these prior methods with local feature-based adaptivity.

**Weaknesses:**

To start, the proposed method appears to be very computationally demanding compared to the alternatives it is benchmarked against. I could not find a runtime comparison in the pdf manuscript, so in the interests of transparency I’d like to ask for it to be disclosed. Overall, the expansion of the label space into custom “label-distribution” space involves a lot of matrix computations, together with many parallel neural networks with non-trivial architecture.

Furthermore, it appears that due to the heavy parameterization of the method, there is a risk of non-robustness to misspecification that could be more pronounced than with the other methods. Thus, it would have been important to see how well this method does for noisy/drift-prone settings where the decision maps and/or labels could be misspecified.

Also, examining the plots, I would agree that the proposed method does exhibit substantial regret benefit over the benchmark methods on aggregate (as well as that the proposed method is more robustly performant, with box widths smaller than the rest), but I would be more moderate in making the performance gains claims given that there is still substantial overlap between the boxes in most cases. Thus, what we can deduce with a lot of certainty is that the proposed method obtains much better regret than the naive benchmark in almost all evaluations, which other methods by and large cannot consistently achieve in the sense of box-plots. However, what I believe we cannot claim with absolute certainty is the superiority of the proposed method over all of the benchmarks at once: For instance, the KNN based benchmark is usually in the same ballpark. Moreover, on real-world multi-item newsvendor (Figure 3), the performance of most methods looks quite evenly matched, modulo the variation/box width.

As another meta-issue, while I appreciated the logical nature of the proposed pipeline, I was not as convinced about the variety of heuristic choices that went into it at most junctures (many of these choices are not ablated against and would in fact be difficult to ablate). This relates to neural net architectures, hyperparameters like the neighbor count M when deciding on the largest transfer components; and this also relates to other subtler design choices that could be made, but were not made and weren’t usually discussed. Just as an example, when finding the M highest transfer cost samples, the hyperparameter M first of all sounds like the performance could be quite sensitive to it; so it appears that similar samples could be clustered together at first before performing this step, as the optimal M would then be found in a smaller, more robust range.

**Questions:**

Please see above. Furthermore, I have some additional questions, which if the authors are able to address them would likely require some different plots from the ones displayed.

First, the adaptively chosen support is mentioned quite a few times, but there are no illustrations that specifically showcase the adaptivity/variation in support throughout the instances on any of the tasks, so I’d request for this to be provided.

Second, there is a lot of mention of the difficulties, related to non-differentiability, of the standard existing predict-then-optimize approaches that are based on designing customized decision-aware losses. Yet, the comparison in the experimental section remains high-level, and doesn’t focus on exhibiting the favorable contrast as it specifically pertains to non-smoothness issues: I could imagine a dataset where decisions are intentionally set to be very discontinuous, and showing the benefits that the current framework has over decision-loss-based ones, locally. Currently, based on the results of this paper, it appears that the added benefit may be in the extra stability of the proposed method, as I imagine it to be quite computationally demanding compared to any decision-loss-optimizing method, smooth or nonsmooth.

Second, returning to the point of the KNN method being one of the most closely matched, this raises the question as to whether the feature-level similarity may in any way have translated to decision-level similarity. If is there a way to display whether that is or is not the case empirically, that would be great; else, a qualitative discussion in the case of each of the two studied settings would suffice.

---

> ### Author Response · Authors · 2025-11-23
>
> **Response to weakness 1**
>
> We appreciate the reviewer’s insightful comment regarding the computational demands of our method. In response, we have added the runtime comparison results in Appendix F of the revised manuscript. While we acknowledge that our approach has a longer training time than baseline models, we argue that this one-time cost does not significantly impact practical use in contextual stochastic optimization (CSO) problems. Additionally, our method offers clear advantages in decision quality and inference time, making it well-suited for real-world applications.
>
> The total computation time consists of training and inference. The training stage is a one-time cost, while inference involves repeated predictions and decision optimization. Our empirical results show that while the training time is longer due to label-space enhancement and richer decision-related information, this enables significantly stronger decision performance, particularly in low-data regimes.
>
> During the inference stage, our method demonstrates a clear advantage. Unlike methods like SAA and KNN, which experience increased inference time as the sample size grows, our approach only requires a single forward pass through the neural network, with the complexity of evaluating the objective function remaining constant. This ensures that inference time remains stable and low, even as sample size increases—an important consideration for real-world applications where inference and decision-making are performed repeatedly.
>
> Thus, although the training time is longer, our method achieves faster inference and superior decision quality, making the overall computational profile highly beneficial for practical decision-making applications.
>
> **Response to weakness 2**
>
> We appreciate the reviewer's concern regarding robustness and address it from three perspectives: parameterization, misspecification, and empirical stability. First, regarding parameterization, we clarify that the distribution weights $\boldsymbol{\pi}_i$ are auxiliary variables used solely during the training phase for Label Enhancement, not as learnable parameters for inference. Far from increasing overfitting risks, the objective in Eq. (10) acts as a regularizer—functioning like a low-pass filter—on the label graph. Therefore, the concern of "heavy parameterization" does not apply to the final deployed model.
>
> Regarding potential misspecification, our method is designed to be distribution-agnostic. Unlike approaches that assume rigid priors (e.g., specific Gaussian families), we utilize a flexible discrete function form capable of approximating arbitrary distributions. This design allows the model to adapt to the underlying data structure without forcing potentially incorrect assumptions, thereby significantly mitigating the risk of misspecification.
>
> Finally, we provide strong empirical evidence of robustness. We have added a sensitivity analysis in Appendix G, demonstrating that our method maintains consistent performance across a wide range of hyperparameters. Furthermore, experiments with artificially injected noise (Section 5) confirm that our decision-aware mechanism effectively filters misleading signals. This resilience is further validated by our superior performance on inherently noisy real-world datasets, proving the method's robustness against data irregularities.
>
> **Response to weakness 3**
>
> We sincerely appreciate the reviewer's careful examination of the experimental results. We have revised the "Contributions" section to be more precise. We shifted the claim from "outperform" to "consistent regret reduction." We acknowledge that the primary advantage of our approach is its reliability in minimizing high-cost errors rather than achieving a drastic margin of victory in simple scenarios.
>
> To better quantify this "relative superiority" amidst the overlapping distributions, we have included a detailed statistical analysis in Appendix E, including the stochastic information of regret of the metrix of regret rank. While box plots show the aggregate distribution of raw values (which can be heavily influenced by the scale of specific instances), the average rank assesses how the method performs relative to competitors on each individual test sample. This is defined as
>
> $$\text{Rank}\_{\text{avg}}(M) = \frac{1}{N} \sum_{i=1}^{N} \text{Rank}_{i}(M)$$
>
> where $N$ is the total number of test samples and $\text{Rank}_i(M)$ denotes the rank of method $M$ on the $i$-th sample among all competing methods (where rank 1 indicates the lowest regret). Crucially, this metric effectively mitigates the skewing effect of extreme values. This analysis indicates that even amidst overlapping regret distributions, our method demonstrates a superior ranking profile, achieving the lowest Average Rank in the majority of experimental settings.

---

> > ### Author Response · Authors · 2025-11-23
> >
> > **Response to weakness 4**
> >
> > Thank you for your valuable comment. We agree that several heuristic choices in our pipeline, including the neural network architecture and the hyperparameter M, deserve further clarification. In response, we have provided detailed disclosures of our experimental settings in Appendix C and conducted additional sensitivity experiments regarding hyperparameter settings in the revised manuscript (see Appendix G). The results show that our method is not highly sensitive to M within a reasonable range, suggesting that model performance is robust to this choice.
> >
> > We also appreciate the suggestion of clustering similar samples before selecting the top-M transfer-cost samples. While this extension is beyond the current scope, we now mention it as a potential future direction of the method.
> >
> > **Response to question 1**
> >
> > We thank the reviewer for this helpful suggestion. In Appendix I, we have added a detailed discussion analyzing the mechanism of our adaptive support compared to fixed discretization strategies. We clarify that the "adaptively chosen support" in our framework operates on a fundamentally different paradigm than fixed support construction, making a direct side-by-side visual comparison challenging. The core advantage of our approach lies in solving the dimension-accuracy trade-off. For continuous target variables, fixed supports require rigid discretization. High accuracy requires fine granularity (e.g., 1000 points for a range of [1, 1000]), leading to prohibitively high prediction dimensionality. Conversely, reducing the number of points sacrifices distributional resolution. Our mechanism circumvents this by dynamically concentrating support points where the probability mass is highest for each specific instance. This allows us to achieve high distributional resolution without the computational burden of a dense, pre-defined grid. The new section in Appendix I details this point.
> >
> > **Response to question 2**
> >
> > Thank you for this constructive comment. The non-differentiability of decision-aware loss functions indeed complicates the training of predictive models. We would like to emphasize that the difficulty is not only in the longer training time but also in whether the resulting estimator can achieve good decision quality. Our approach leverages label distribution learning to cleverly incorporate downstream decision-making during the label enhancement stage, thereby avoiding the issues caused by non-differentiability.
> >
> > From our experiments, while our method incurs a longer training time, it shows significant advantages in both decision quality and inference time. Given that training is done only once in practical applications, we believe the trade-off between training time and the improved decision quality and inference speed is reasonable and beneficial.
> >
> > Additionally, we wish to clarify that the non-differentiability in decision-loss optimization stems from the nature of the downstream decision problem, rather than the dataset itself (Bertsimas and Kallus, 2020; Elmachtoub and Grigas, 2022). For instance, in linear programming, the optimal solution invariably lies at an extreme point. As uncertainty parameters vary, the optimal solution can jump abruptly between extreme points, rendering the regret function non-differentiable.
> >
> > Nevertheless, to further address your concern, we have included an additional discrete optimization task in Appendix D. Furthermore, we emphasize that the quadratic cost network flow problem in our main experiments also entails discrete optimization. Together, these distinct tasks demonstrate our method's robustness in complex, non-smooth decision environments.
> >
> > **Response to question 3**
> > Thank you for raising this question. Our method is not conceptually as close to KNN as might initially appear, and feature-level similarity does not generally translate into decision-level similarity from a theoretical perspective. In many decision problems, the mapping from labels to decisions is often nonlinear and asymmetric. For example, in the Newsvendor problem, if the true demand is $y=5$, the predictions $3$ and $7$ are equidistant in feature space ($|5-3|=|5-7|$). However, they incur vastly different decision regrets because understocking and overstocking costs are rarely equal. Thus, minimizing feature distance (as KNN does) fails to minimize decision regret.
> >
> > To validate this disconnect, we added a heatmap visualization in Appendix H. This comparison reveals that the structure of a standard feature-based similarity matrix differs significantly from our decision-aware matrix.
> >
> > Reference:
> >
> > Bertsimas, D., Kallus, N., 2020. From predictive to prescriptive analytics. Management Science 66, 1025–1044. https://doi.org/10.1287/mnsc.2018.3253
> >
> > Elmachtoub, A.N., Grigas, P., 2022. Smart “predict, then optimize.” Management Science 68, 9–26. https://doi.org/10.1287/mnsc.2020.3922

---

### Author Response · Authors · 2025-11-23
**Rebuttal Summary**

We sincerely thank the reviewers for their constructive comments, which have significantly improved the quality and clarity of our work. Based on your suggestions, we have extensively revised the manuscript. The major updates are summarized below:

* **Methodological Clarifications:** We added **Appendix B** (theoretical analysis of the "point-mass" edge case), **Appendix C** (detailed experimental protocols), and **Appendix I** (illustrative example of adaptive support). We also adopted a **strict chronological split** in experiments to prevent temporal leakage.

* **New Discrete Decision Task:** To verify applicability in non-smooth environments, we introduced the **Discrete Newsvendor Problem** in **Appendix D**. Together with the quadratic cost network flow problem (also discrete), these experiments confirm our method's robustness in discrete optimization.

* **Quantitative Evaluation:** We expanded **Appendix E** to include detailed tabular data (Mean $\pm$ Std) and a **ranking-based evaluation**, confirming the consistent superiority of our method.

* **Computational Efficiency:** We added a detailed runtime analysis (training vs. inference) in **Appendix F** to clarify trade-offs and justify the offline training cost.

* **Hyperparameter Robustness:** We included a comprehensive sensitivity analysis (specifically for $M$ and $\alpha$) and a discussion on parameter selection in **Appendix G**.

* **Interpretability:** We added **Appendix H** to visualize the decision-aware similarity matrix ($S$) against a feature-based baseline, illustrating how our method captures decision-level structures.

Detailed point-by-point responses follow below.

---

### Meta-Review · Area_Chair_Sw9A · 2026-01-13

**Summary:**

This paper proposes a framework for contextual stochastic optimization that encodes decision-awareness at the data representation level rather than through custom loss functions. The framework transforms scalar targets into discrete mixture distributions using decision-aware similarity matrices derived from optimization transfer costs, then trains dual-branch neural networks to predict these distributions. The authors demonstrate regret reduction on synthetic and real-world benchmarks based on classical optimization problems.

This work received lukewarm feedback from reviewers in their initial reviews: scores of (4, 4, 4, 4) with moderate to low confidence. In general, most reviewers found the approach novel and the pipeline intuitive. The paper also demonstrated the viability of the method through consistent empirical performance on classical optimization problems. In this case, the low scores mapped to a number of disparate concerns about significance of the work—e.g., supporting design choices, studying its limitations, establishing where it could or could not be useful.

Having read the reviews, the rebuttal, and the paper, I am unfortunately recommending rejection. I want to be clear that this was not an easy decision. My decision reflects concerns about exposition and completeness rather than the core idea. I see the borderline scores and low confidence scores as license for some adjudication from an AC. On my end, I find the core idea of the paper interesting and potentially valuable to the machine learning community. My concern is that the paper – following the reviews – does not yet provide the information needed to evaluate the significance and soundness to its reviewers. Even as these issues could be addressed for a potential camera ready, I believe that the changes would deserve another round of peer review. Beyond the concerns voiced by reviewers, I believe the work would benefit from a more detailed analysis explaining how the ideas could be extended, along with a frank discussion of benefits and limitations. At a venue like ICLR, I think the work would benefit from clearer exposition targeting a broader audience (e.g., a gentle introduction to the problem that lists concrete use cases and an accessible Figure 1).

**Reviewer Concerns:**

1. Limited Experimental Scope (only two problem domains tested) (vKBN, nUpX, FvXG)
4. Exposition and Accessibility (heuristic choices not explained, low reviewer confidence) (vKBN, 3Qg6)
2. Missing Baselines (no comparison with end-to-end decision learning methods) (3Qg6, FvXG)
3. No Theoretical Guarantees (formal justification for when method succeeds) (nUpX)

**Reviewer Scores:**

1-2 point increase around the board.

---

### Decision · Program_Chairs · 2026-01-26

Reject